# Limited influence of irrigation on pre-monsoon heat stress in the Indo-Gangetic Plain

Roshan Jha[1], Arpita Mondal [1,2✉], Anjana Devanand [3,4], M. K. Roxy [5] & Subimal Ghosh[1,2]

Hot extremes are anticipated to be more frequent and more intense under climate change, making the Indo-Gangetic Plain of India, with a 400 million population, vulnerable to heat stress. Recent studies suggest that irrigation has significant cooling and moistening effects over this region. While large-scale irrigation is prevalent in the Indo-Gangetic Plain during the two major cropping seasons, Kharif (Jun-Sep) and Rabi (Nov-Feb), hot extremes are reported in the pre-monsoon months (Apr-May) when irrigation activities are minimal. Here, using observed irrigation data and regional climate model simulations, we show that irrigation effects on heat stress during pre-monsoon are 4.9 times overestimated with model-simulated irrigation as prescribed in previous studies. We find that irrigation increases relative humidity by only 2.5%, indicating that irrigation is a non-crucial factor enhancing the moist heat stress. On the other hand, we detect causal effects of aerosol abundance on the daytime land surface temperature. Our study highlights the need to consider actual irrigation data in testing model-driven hypotheses related to the land-atmosphere feedback driven by human water management.

---

[1] IDP in Climate Studies, Indian Institute of Technology Bombay, Mumbai 400076, India. [2] Department of Civil Engineering, Indian Institute of Technology Bombay, Mumbai 400076, India. [3] Australian Research Council Centre of Excellence for Climate Extremes, University of New South Wales, Sydney, NSW, Australia. [4] Climate Change Research Centre, University of New South Wales, Sydney, NSW, Australia. [5] Centre for Climate Change Research, Indian Institute of Tropical Meteorology, Pune 411008, India. ✉email: marpita@iitb.ac.in

Human water management activities such as irrigation can mitigate hot and dry extremes[1–3]. The biogeophysical impacts of irrigation on hot extremes are well known: irrigated areas have high land evaporation under hot and dry conditions, leading to reduced land surface temperature and atmospheric moisture deficit[4]. Modelling studies in the past have revealed varying magnitudes of cooling effects of irrigation, accounted for as soil moisture at field capacity[5,6] or as a percentage of soil saturation[7,8] or from a water balance model[9–11]. Multiple studies[2,12] also showed the magnitude of cooling is a linear function of the volume of irrigation through sensitivity analyses. However, non-irrigated areas under dry conditions favour the hot extremes through a rise in sensible heat and reduced evapotranspiration, causing dry heat stress on the human body[13,14]. The role of soil moisture-temperature coupling in developing hot extremes is well-documented[15,16].

India experiences hot extremes resulting in human mortality and agricultural crop failures during the pre-monsoon season (April–May) and early monsoon season, in June[17–19]. Observational studies point to persistent anti-cyclones with depleted soil moisture during pre-monsoon as the major factor for producing and amplifying hot extremes in India[20–22]. Other studies[12,17,23] also discuss the confounding role of human water management practices such as irrigation on hot extremes over the Indo-Gangetic Plain. Indo-Gangetic Plain is a highly irrigated region with a population density of more than 1000 per sq. km, experiencing intensifying hot and humid weather associated with hot extremes under global warming[24,25]. Hot extremes under humid conditions can reduce the human body capacity to maintain a healthy body temperature by perspiration, thereby inducing moist heat stress. Therefore, it is imperative to adequately quantify the irrigation feedback on dry heat stress and moist heat stress given by dry-bulb temperature and wet-bulb temperature, respectively, in this region. Further, for data-scarce regions such as the Indo-Gangetic Plain, where irrigation practices are unique, it is also necessary to ensure that the volume and area of irrigation estimated in the model represent the actual field conditions.

In India, large-scale irrigation is observed only during the monsoon and post-monsoon seasons: Kharif and Rabi, extending from June to September and November to February, respectively. The rest of the months, March to May, are usually hot and dry without extensive agricultural activities because of cropping pattern[26,27] and government policies related to groundwater conservation[28]. Heat stress associated with hot extremes is observed during the pre-monsoon season (April and May) and in the early monsoon season (June), specifically in the late monsoon onset years with high-irrigation feedback from the monsoon (Kharif season) irrigation practices. Recent studies[3,12] on irrigation feedback during the pre-monsoon season used land surface models to estimate irrigation amounts in the absence of region-specific irrigation data over the Indian region. Further, these studies used annual irrigated areas instead of pre-monsoon seasonal area fractions, failing to account that agricultural activities and irrigation in the field are minimal during pre-monsoon hot extremes. While such approaches may be suitable for other regions globally, they may overestimate pre-monsoon irrigation amounts in the Indo-Gangetic Plain. Hence, the model-estimated irrigation volumes lead to very high feedback on near-surface climate compared to changes in observed records of land surface temperature and wet-bulb temperature. Therefore, attributing decreasing land surface temperature[12,17] and rising wet-bulb temperature[12,23,25] over the Indo-Gangetic Plain to irrigation[12] alone based on overestimated irrigation overlook contribution from other factors like aerosol loading[29] and remote moisture transport as important as irrigation.

In this work, using satellite observations and prescribing the seasonal irrigation data from India's agricultural census[26] in the coupled Weather Research and Forecasting-Community Land Model, we show that irrigation has a limited role in hot extremes and associated heat stress during the pre-monsoon season. Here, the dry heat and moist hot stresses are represented by the daily dry-bulb temperature and wet-bulb temperature, respectively, and their extremes using the 95th percentile of their representing variable. Further, we demonstrate that decreasing land surface temperature over the Indo-Gangetic Plain has a stronger causal relationship with high aerosol loading in the region than evapotranspiration. We also report that increasing humid conditions during pre-monsoon can be due to non-local moisture that may not be associated with irrigation activities.

## Results

**Pre-monsoon temperature extremes and non-cropping season.** The daily maximum temperature from Indian Meteorological Department (IMD) averaged from 1981 to 2020 over the Indo-Gangetic Plain, reaches the peak in May; however, the monthly enhanced vegetation index (EVI), an improved measure of vegetation density, averaged over the available period 2001–2020 shows minimum values during May (Supplementary Fig. 1). Further, the dip in temperature with an increase in EVI in June depicts the onset of the Indian summer monsoon and the beginning of the major cropping season (Kharif season). Monsoon season/Kharif season observes extensive irrigation compared to pre-monsoon (April and May), suggesting the contrasting irrigation feedback on hot extremes during the two seasons. Thus, it can be understood that high temperature favouring the hot extremes in pre-monsoon coincides with minimum agricultural activities and limited irrigation, as cropland is the major land-use land cover class over the Indo-Gangetic Plain (Supplementary Fig. 2), need to be studied using region-specific data and a region-specific model. To substantiate this, we use the agricultural census-based data from the Government of India and satellite observations to understand the cropping patterns and vegetation response during pre-monsoon and monsoon seasons. The annual irrigation area fraction over India based on the Food and Agricultural Organization (FAO) Global Map of Irrigated Areas (GMIA)[30] (Fig. 1a) and monsoon season irrigation fraction[31] (Fig. 1c) from the agricultural census data over the Indo-Gangetic Plain shows that more than 80% of the Indo-Gangetic Plain region under cultivation is irrigated. The lower irrigation area fraction values during the pre-monsoon season (Fig. 1b), except for a few sugarcane growing hotspots, indicate that pre-monsoon is a non-cropping season. In addition, the irrigation water requirement for paddy and non-paddy crops taken from Fishman et al.[32] shows lower irrigation water for paddy and non-paddy crops during the pre-monsoon season than monsoon season (Supplementary Fig. 3). Spatial plot of EVI and evapotranspiration (ET) observations from the MODIS sensor from NASA's Earth Observing System over 2001–2020 during the pre-monsoon and monsoon season support the agricultural census irrigation fraction data over the Indo-Gangetic Plain. The dominance of vegetation during monsoon indicated by high EVI values of 0.4-0.6 compared to low values of 0.1-0.2 during pre-monsoon season over croplands of Indo-Gangetic Plain re-establish monsoon as a major cropping season, as is widely known in India (Fig. 1d, e). Correspondingly, the Indo-Gangetic Plain shows minimal ET during the pre-monsoon season compared to the monsoon season (Fig. 1f, g), depicting very limited vegetation and soil moisture. Another vegetation metric, the MODIS-derived leaf area index (LAI), further supports this finding (Supplementary Fig. 4). With the absence of extensive irrigation activities, dry and

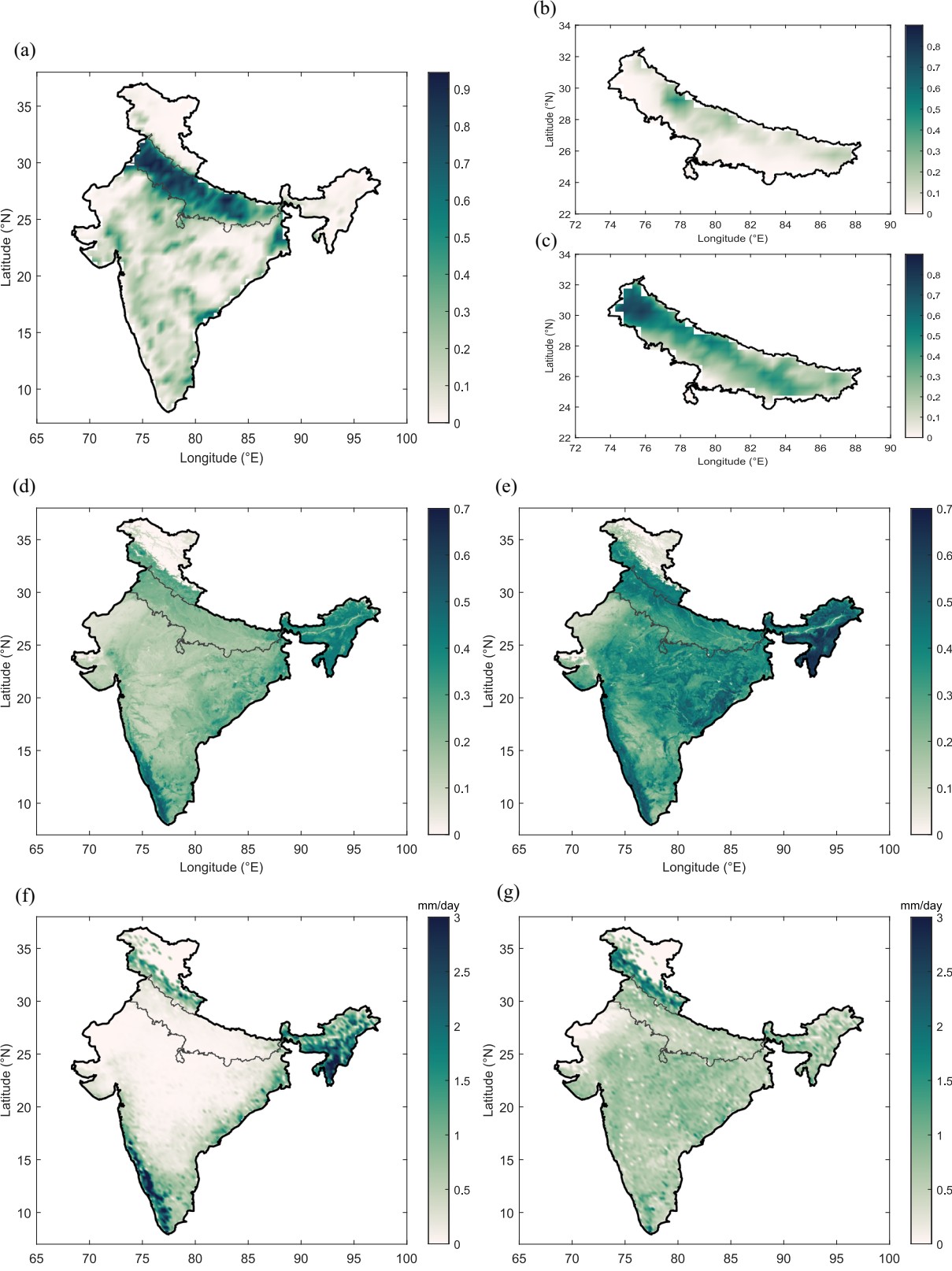

**Fig. 1 Comparison of pre-monsoon (April–May) and monsoon (June–September) in India. a** FAO annual irrigation fraction over India; Mean
(2004–2016) fraction of grid cell under irrigation including paddy and non-paddy crop from agricultural census-based data during **b** Pre-monsoon
**c** Monsoon; Mean EVI for the period 2001–2020 from moderate resolution imaging spectroradiometer (MODIS) for **d** pre-monsoon **e** monsoon; Mean ET
(mm/day) for the period 2001–2020 from MODIS for **f** pre-monsoon **g** monsoon.

uncropped lands during pre-monsoon, in fact, have high surface temperature and more potential to develop hot extremes than urban environment as suggested by negative surface urban heat island effect during daytime over the Indo-Gangetic plain[13,33]. Therefore, using global irrigation datasets or other proxies of irrigation[12] that do not consider such seasonal variability of irrigation activities in the field may lead to flawed conclusions over the regions like Indo-Gangetic Plain during April–May.

**Irrigation feedback on heat stress.** Previous model-driven studies[1,3,12] show that irrigation over the Indo-Gangetic Plain has the potential to decrease pre-monsoon dry heat by surface cooling and increase pre-monsoon moist heat through irrigation-induced moisture. These studies showed meteorological response to model-estimated irrigation volume based on annual averages instead of realistic irrigation estimates for the pre-monsoon season. Here, we prescribe both the actual and model-estimated irrigation volume in coupled Weather Research and Forecasting-Community Land Modelv4 (WRF-CLM4) to investigate the impact of realistic irrigation compared to model-estimated irrigation. We performed simulations for the pre-monsoon summer (April–May) at a 30 km spatial resolution for the period 2004–2016 with three scenarios, namely CTL (no irrigation), AGR (irrigation based on agricultural census data), and MOD (model-estimated irrigation) using initial and lateral boundary conditions from European Centre for Medium-Range Weather Forecast Interim Re-Analysis. The AGR simulation uses a combination of irrigation water estimates based on agricultural census over the Indo-Gangetic Plain and monthly global irrigation water use estimate[34] over the rest of India. The MOD experiment estimates irrigation water over GMIA areas based on soil moisture deficit with target soil moisture set to field capacity of the soil[12]. We note that this is seldom the case in reality in India, even during the main irrigation season[31]. Model-simulated 2m-air temperature from three different parameterisation combinations are compared with those from Indian Meteorological Department (IMD) observed data[35] to assess model skill over India. The WRF-CLM4 model with MYNN3-WSM6-Grell 3D combination shows the least difference between observed and simulated results over Indo-Gangetic Plain for April–May during 2004 and 2006 (Supplementary Fig. 5). Moreover, the pattern correlation for two variables: daily mean temperature and daily maximum temperature between model simulated and observed data showed a correlation value of greater than 0.6 over the Indo-Gangetic Plain (Supplementary Fig. 6).

We compute the mean land surface temperature (LST) differences over the simulation period between three experiments (CTL, MOD, and AGR). The difference between MOD and CTL exhibits large cooling of about 2.07 °C averaged over Indo-Gangetic Plain in response to model-estimated irrigation (Fig. 2a), in agreement with earlier studies[12]. In contrast, the difference between AGR and CTL shows the limited irrigation cooling effect of 0.42 °C averaged over the same area (Fig. 2b). Finally, the difference between MOD and AGR provides the magnitude by which the cooling effect is overestimated with model-estimated irrigation data compared to realistic irrigation data (Fig. 2c). The influence of model-estimated irrigation is 4.9 times higher than the census-based irrigation on land surface temperature, with the mean overestimation value of 1.65 °C spatially averaged over the Indo-Gangetic Plain. Similarly, the moistening effect is also overestimated by model-estimated irrigation application over Indo-Gangetic Plain as census-based irrigation reduces relative humidity by only 2.47%, around 10% lesser than model-estimated irrigation (Supplementary Fig. 7). The high temperature during April–May results in soil moisture drying, leading to

high-irrigation demand over higher area fractions in MOD simulations. However, in reality, the irrigated area fraction and irrigation water use are limited in most parts of the Indo-Gangetic plain during the pre-monsoon summer, as presented by AGR simulations. Their differences present the overestimation of irrigation feedback to the cooling and moistening. Similar high feedback from model-estimated irrigation is visible in our results to other meteorological variables: maximum air temperature, mean air temperature, specific humidity, and wet-bulb temperature (Supplementary Fig. 8) because of lack of accounting for realistic irrigation practices in the model simulations.

Moreover, the results from another set of simulation named as HNG using Huang et al.[34] monthly irrigation withdrawal data all over India for four years (as described in Supplementary Note 2) shows high feedback to meteorological variables (Supplementary Fig. 9) similar to the MOD experiment over the Indo-Gangetic Plain. The most probable reason is the tendency of water models to estimate higher irrigation water for dry soil conditions over the annual irrigation area fraction given by GMIA data during the pre-monsoon season. The agricultural census-based irrigation volume prescribed to the model (ARG) overcomes this drawback and shows the actual influence of irrigation.

To understand the feedback from the census-based irrigation volumes to the dry and moist heat stress, hereafter, we focus only on the Indo-Gangetic Plain. We compute daily maximum temperature (Tmax) and daily wet-bulb temperature (Tw) to represent dry heat and moist heat. Irrigation has resulted in only a 0.55 °C drop in maximum temperature, decreasing by only 1.5% (Supplementary Table 2 and Fig. 3a). Likewise, irrigation influence on daily mean temperature (T2) is even lower, cooling of 0.4 °C over the Indo-Gangetic Plain (Fig. 3b and Supplementary Table 2). However, model-estimated irrigation (MOD) overestimate daily maximum and mean temperature by four times (Supplementary Fig. 8a, b). Overall, dry heat appears to have limited cooling by irrigation activities over Indo-Gangetic Plain except for some sugarcane growing hotspot areas in Uttar Pradesh, as small changes in latent heat and sensible heat (Supplementary Fig. 10a, b) fail to bring substantial change in air temperature. These results contrast with earlier studies[2,12] that have demonstrated 4–8 °C cooling over India attributed to irrigation effects. Second, we find irrigation has made a small contribution to the increment in specific humidity and wet-bulb temperature (Fig. 3c, d). Limited irrigation during pre-monsoon supplied the moisture to some extent to the evaporative demand created by high atmospheric moisture deficit with clear skies. Further, the difference in specific humidity (0.0005 kg/kg) between AGR and CTL over all land areas of the Indo-Gangetic Plain (Supplementary Table 2) is four and half times lesser than the response obtained from MOD simulations. In addition, actual irrigation increased wet-bulb temperature by just 0.27 °C, proving that irrigation-induced local moisture has a very limited role in raising moist heat stress. Moreover, the sensitivity test of daily mean temperature (Supplementary Fig. 11), daily maximum temperature (Supplementary Fig. 12), and wet-bulb temperature (Supplementary Fig. 13) response to model-estimated irrigation and agricultural census-based irrigation for a different combination of parameterisation schemes (Supplementary Note 3) show results are quite robust. The error bar diagram (Supplementary Fig. 14) shows that the agricultural census-based irrigation has a similar influence with three parameterisation combinations.

**Irrigation feedback on heat stress extremes.** Further, we investigate the response of dry and moist heat extremes to realistic irrigation volumes using the AGR and CTL simulations (Supplementary Fig. 15). The 95th percentile of daily maximum temperature

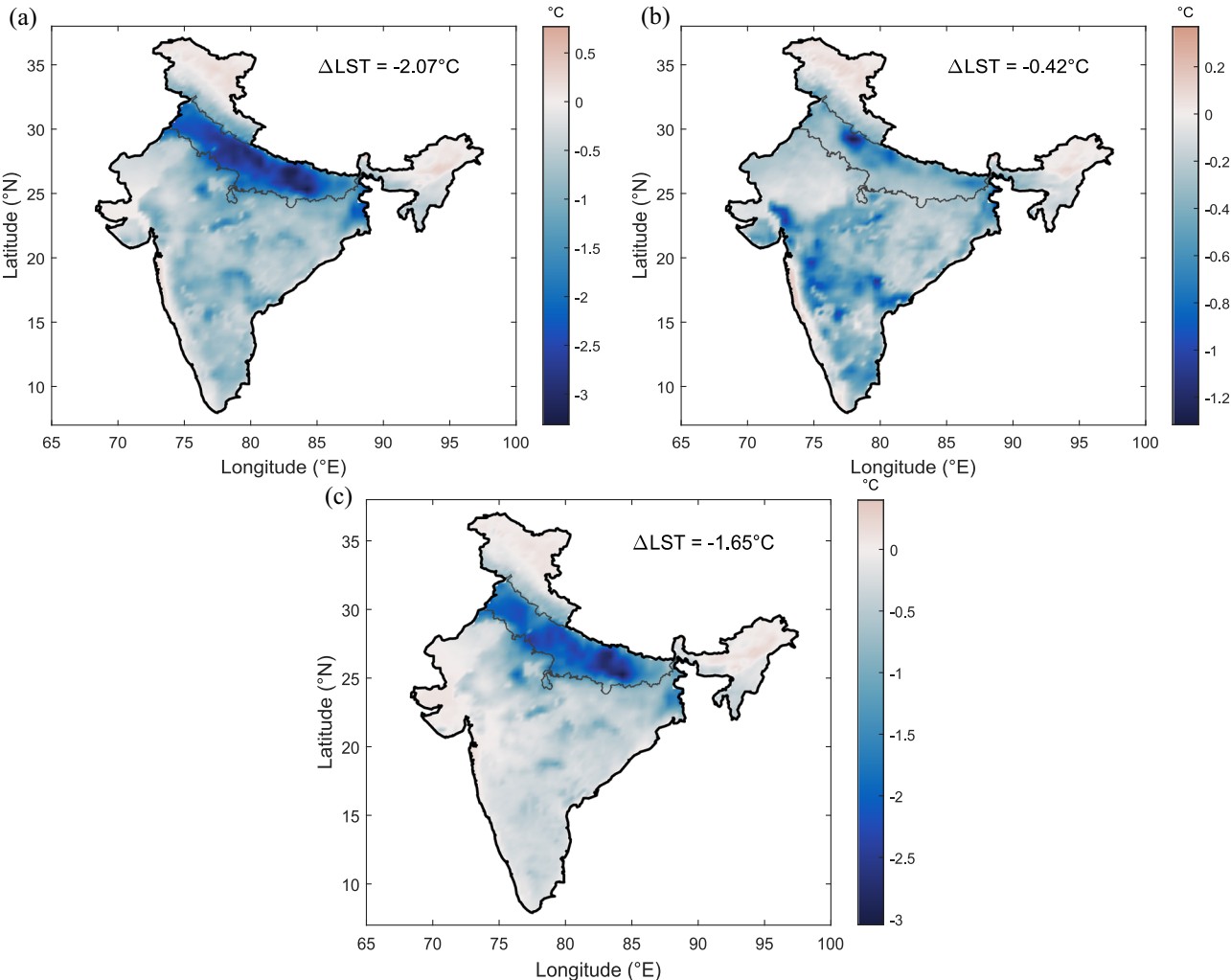

**Fig. 2 Influence of model-estimated and agricultural census-based irrigation volume on land surface temperature (LST).** Mean (2004–2016) difference in land surface temperature (°C) between different experiments. **a** Influence of model-estimated Irrigation (MOD—CTL). **b** Influence of agricultural census irrigation (AGR—CTL). **c** Magnitude of overestimation of irrigation feedback (MOD—AGR). The mean difference spatially averaged over Indo-Gangetic Plain is shown as ΔLST. CTL represents WRF-CLM4 simulation with no irrigation, AGR represents WRF-CLM4 simulation with agricultural census-based irrigation data and MOD represents WRF-CLM4 simulation with model-estimated irrigation data.

(Tmax_95) and daily wet-bulb temperature (Tw_95) during the pre-monsoon season represent extreme dry heat and moist heat conditions, respectively. The difference in average Tmax_95 during 2004–2016 in AGR and CTL scenarios shows that irrigation reduces extreme dry heat by less than 1 °C (Supplementary Fig. 15a) in contrast with upto 3 °C reductions during annual maximum daytime temperature shown in model-driven study[1]. Extreme moist heat conditions represented by Tw_95 have increased by 0.22 °C over Indo-Gangetic Plain with irrigation application, which is lower than the previous model-simulated estimates of 3–6 °C shown in the model-driven study[12] (Supplementary Fig. 15b). Here, the extreme dry heat and moist heat response to model-estimated irrigation are higher than census-based irrigation and consistent with previous model-driven studies (Supplementary Fig. 16). Further, irrigation prescribed in the AGR experiment does not substantially reduce the planetary boundary layer (PBL) height compared to the MOD experiment (Supplementary Fig. 17a, b).

**Role of aerosols and non-local moisture.** With observed and model-simulated data proving the limited role of irrigation on pre-monsoon heat stress in the region, we further investigate the possible role of two more important causal factors—aerosols and

non-local moisture. The spatial variation of mean aerosol optical depth (AOD) from MODIS aerosol products during the pre-monsoon season over India from 2001 to 2020 is shown in Fig. 4a. The aerosol loading over the Indo-Gangetic Plain is higher than the rest of India, with AOD values of about 0.5–0.9. Multiple inventories have shown abundance and increasing trends of anthropogenic and biomass burning emissions over India for 1980–2010, contrary to other regions such as Europe and the USA[36]. The densely populated Indo-Gangetic plain is a primary source of aerosol emission in India, reportedly showing high values of AOD[37–39].

The MODIS daytime LST data over the same period shows lower land surface temperature over the Indo-Gangetic Plain compared to the rest of India (Fig. 4b). Anthropogenic and natural aerosols can change surface temperature through the change in radiative energy budget by scattering and absorption of incoming radiation[40,41]. Model-driven studies have shown the direct and indirect effects of aerosol-radiation interaction on lowering the temperature[42] and increasing the relative humidity[29]. Notwithstanding the fact that the region is also marked with absorbing aerosols that can exacerbate dry hot extremes[43], we further examine the relationship between temperature and aerosols over

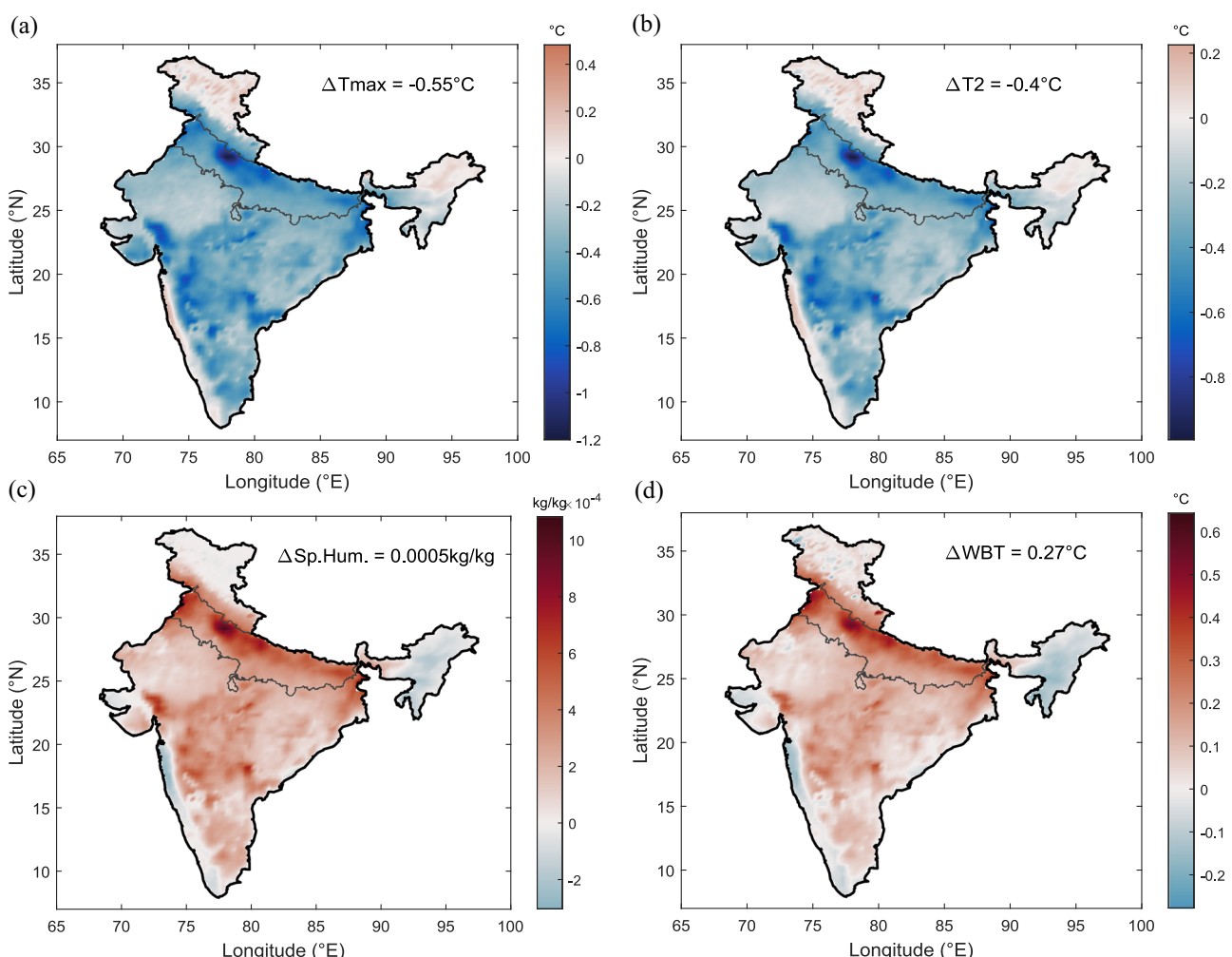

**Fig. 3 Influence of agricultural census-based irrigation on air temperature, specific humidity, and wet-bulb temperature (Tw).** Difference between AGR and CTL experiment during pre-monsoon season (April–May) for the period 2004–2016 for **a** Daily maximum temperature (°C). **b** Daily mean temperature (°C). **c** Specific humidity (kg/kg). **d** Wet-bulb temperature (°C). CTL represents WRF-CLM4 simulation with no irrigation and AGR represents WRF-CLM4 simulation with agricultural census-based irrigation data.

Indo-Gangetic Plain (Fig. 4c). A significant negative linear relationship is observed between AOD and daytime LST ($r = -0.46$, $p = 1.722 \times 10^{-41}$). However, a weak positive linear relationship is witnessed between AOD and daytime LST over the Indian region as AOD and LST observed over the rest of India are different from the Indo-Gangetic plain (Supplementary Fig. 18a). We also perform the statistical Granger Causality test on the satellite observations of daytime LST and AOD from MODIS (Fig. 4d). The results showed a statistically significant causal relationship; AOD Granger causes LST, attributing a spatially prominent decrease of LST over Uttar Pradesh and Bihar during April and May to high aerosol concentrations. The same test, when performed with daytime LST and evapotranspiration from MODIS, showed statistically insignificant causal relation between them, indicating that evapotranspiration from irrigated fields does not contribute to the lowering of temperature over the Uttar Pradesh and Bihar (Fig. 4d). However, the results showed a statistically significant causal relationship for both the test: AOD Granger-causes LST and ET Granger-causes LST over Punjab and Haryana showing ET from limited irrigation have impact over temperature (Supplementary Fig. 18b).

Both irrigation activities and aerosol loading have increased over Indo-Gangetic Plain since 1980 or before. However, a significant irrigation expansion is observed only during

monsoon[44], and aerosol loading increasing since the 1980s is high during pre-monsoon[38], which depicts that the cooling and moistening trend that is observed during pre-monsoon is associated with other factors like high aerosol loading[29] than limited pre-monsoon irrigation over Gangetic Plain (Uttar Pradesh and Bihar). In addition, we show a backward air trajectory plot ending at 9:00 UTC (Local time 14:30) for 20 seasonal (April–May) maximum wet-bulb temperature days from 2001–2020 with a source each at Bihar (Supplementary Fig. 19a) and Uttar Pradesh (Supplementary Fig. 19b) using the NOAA HYSPLIT model. The backward air trajectory ending at 50 m above ground level shows that air is transported over the Gangetic Plain region from the ocean (for source at Bihar) and the northern Himalayan range (for source at UP). The moist air coming from these sources during the extreme moist heat condition can be the additional reason for increased moist heat stress over Gangetic Plain. Finally, the exact magnitude of cooling and moistening due to aerosol and non-local moisture over the Indo-Gangetic Plain remains an important factor that can be studied further.

Our results show that modelling studies should use actual irrigation data and account for practices unique to the region rather than generalising schemes or methods originally tailored for other, significantly different hydroclimatic or social

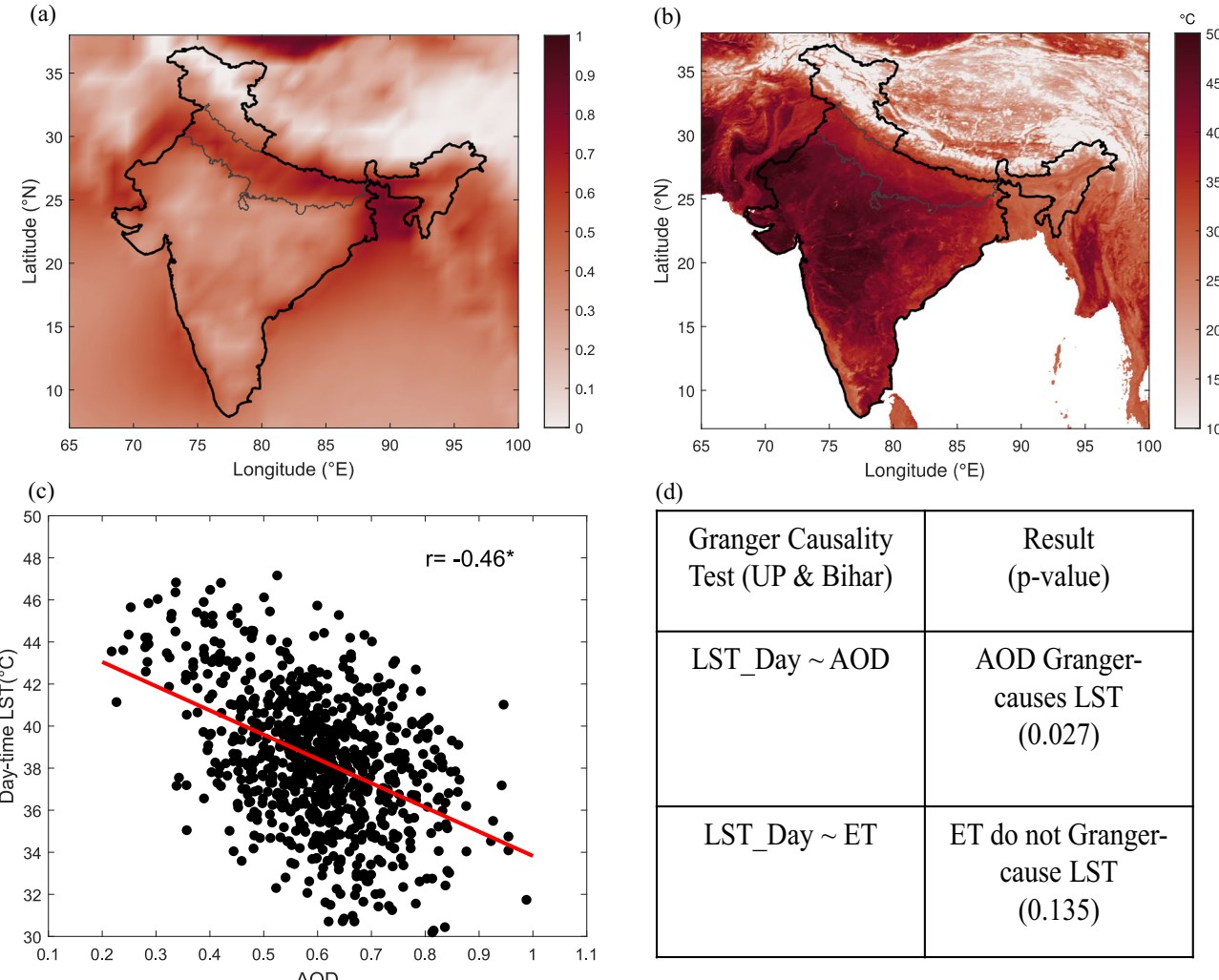

**Fig. 4 Relationship between MODIS aerosol optical depth (AOD) and MODIS daytime LST (°C).** Seasonal mean (April–May) during the period 2001–2020 for **a** AOD, **b** Daytime LST, **c** Scatter plot with Pearson's correlation coefficient (*r*) for the Indo-Gangetic Plain. **d** Causal relationship between variables for Uttar Pradesh (UP) and Bihar: aerosol optical depth (AOD), evapotranspiration (ET) and daytime land surface temperature (LST) from MODIS using Granger causality test in vector auto-regression model (VAR) framework. The null hypothesis for the test is that lagged Variable1 do not explain the variation in Variable2.

conditions. The model-driven hypothesis testing with a state-of-the-art modelling framework is found to be inadequate in simulating the Indian human-natural climate system resulting in a non-realistic conclusion. With the Governmental agricultural census-based irrigation data and regional land-atmosphere model, designed for Indian agricultural practices, we showed that irrigation have limited effect on the moist heat stress to explain the recent trends[12] of wet-bulb temperature and humidity over the Indo-Gangetic basin in the pre-monsoon season. Further, our results do not show strong attenuation of hot extremes by irrigation activities during the pre-monsoon season in the Indo-Gangetic Plain in India and show the possible role of other local factors like aerosol loading in moderating hot extremes. This study also underlines the benefits of the availability of real-field data for model-driven hypothesis testing related to land-atmosphere feedback controlled by human water management. Finally, it can be suggested that any regional representation of a human-natural climate system moderating or amplifying the effects of greenhouse gases needs consideration of regional characteristics and processes; the use of models adopted from different regions may lead to erroneous conclusions.

## Methods

**WRF-CLM simulation**. The Weather Research and Forecasting model[45] coupled to the Community Land Model version 4 (WRF-CLM4) is used for understanding the impact of irrigation on moist and dry heat stresses in India. Three sets of seasonal pre-monsoon (February–May) simulations are performed from 2004 to 2016, with two initial months taken as a spin-up period at the horizontal spatial resolution of 30 km and 30 vertical levels. A similar spin-up period is considered in other studies[31,46,47] for performing experiments over the Indian region in WRF-CLM. All three simulations: CTL (irrigation-off), AGR (irrigation-on with agricultural census-based irrigation data), and MOD (irrigation-on with model-estimated irrigation data), uses initial and lateral boundary conditions from European Centre for Medium-Range Weather Forecast Interim Re-Analysis[48]. The different parameterisation schemes used in the three simulations are MYNN3 Scheme to represent the planetary boundary layer, Grell 3D Cumulus scheme to represent convective parameterisation and the WRF Single Moment 6 class scheme for microphysics. A newer version of the Rapid Radiative Transfer Model (RRTMG) for longwave and Dudhia scheme for shortwave radiation represents radiation schemes, and a revised MM5 surface layer scheme based on Monin–Obukhov Similarity Theory is used for the surface layer scheme. These parameterisation schemes were selected from the three parameterisation combinations described in Supplementary Note 3.

The CTL simulation using WRF-CLM4 is performed without any irrigation module. For the AGR simulation, we employ the WRF-CLM4 model with modules to represent irrigation, groundwater pumping and flooded paddy fields to represent the summer irrigation over parts of the Indo-Gangetic Plain[31]. Additional details on the representation of irrigation, groundwater pumping, and paddy fields in the regional model code can be obtained from Devanand et al.[31]. The agricultural

census-based irrigation estimates are prescribed uniformly at a daily time step during the pre-monsoon season. For each day in the season, irrigation is applied for four hours starting at 6 a.m. local time. Unlike AGR, MOD simulation estimates irrigation water use in the model from the soil moisture deficit, checked at 6 a.m. local time, and applied evenly over four hours each day during the simulation period. Here, the target soil moisture is set to the field capacity of the soil. The soil moisture deficit is the difference between the field capacity of soil and root-zone soil moisture.

**Irrigation data for AGR simulation**. For the AGR simulation, WRF-CLM4 model[31] with irrigation, groundwater pumping, and paddy field module is used to prescribe irrigation using agricultural census-based irrigation estimates for realistic irrigation representation over Indo-Gangetic Plain. The model requires the irrigation data inputs like irrigated area fractions of the grid cells, the proportion of irrigation water used from surface water and groundwater sources, and the irrigation water use over the region, which are estimated using district-wise agricultural census-based data[26] for the Indo-Gangetic Plain (Bihar, Uttar Pradesh, Haryana, Punjab, and Rajasthan). For the rest of India, the irrigation area fraction of the grid cells with source irrigated area is taken from the Food and Agricultural Organisation (FAO) Global Map of Irrigated Areas (GMIA)[49], and irrigation water use estimates based on GMIA estimates are taken from Huang et al.[34]. The description of the preparation of agricultural census-based irrigation data is provided in Supplementary Note 1.

**Model output analysis**. We use IMD[35] daily mean temperature to assess the model skill over the Indo-Gangetic Plain. The simulated meteorological variables (mean temperature, maximum temperature, minimum temperature, specific humidity, relative humidity, pressure, heat fluxes, and wet-bulb temperature) at an hourly scale is converted to daily scale and then averaged over the simulation period for each set of the experiment. The hourly wet-bulb temperature is calculated using hourly mean temperature, hourly dew-point temperature, and hourly surface pressure using the iterative procedure described by Stipanuk[50] available in NCL. The dry heat stress and moist heat stress are represented by daily 2m-air temperature and wet-bulb temperature, similar to the definition adopted for model-simulated output in Mishra et al.[12]. The 95th percentile of daily maximum temperature (Tmax_95) and daily wet-bulb temperature (Tw_95) during the pre-monsoon season represent the extreme dry and moist heat conditions, respectively. The above-mentioned meteorological variables are spatially averaged over Indo-Gangetic Plain for the AGR, MOD, and CTL to understand the impact of census-based irrigation and model-estimated irrigation. The spatially averaged variables are temporally averaged from 2004 to 2016 for all the experiments, and the difference between them quantifies the irrigation feedback due to different irrigation prescriptions. The two-sample *t*-test is also performed on a spatially averaged variable of the AGR experiment to determine whether there is a significant change in values at a daily and annual scale. Here, the null hypothesis for the test is that daily/annual values from AGR and CTL are independent random samples from normal distributions with equal means and equal but unknown variances. The null hypothesis is rejected at the 5% significance level.

**Observed/satellite data**. We obtained the MOD16A2 version-6 8-day composite Evapotranspiration (ET) dataset produced at 500 m resolution to understand the ET response to irrigation during the pre-monsoon and monsoon seasons. The ET dataset is produced using the Penman–Monteith equation and includes the meteorological reanalysis data, and Moderate Resolution Imaging Spectro-radiometer (MODIS) remotely sensed data products such as vegetation property dynamics, albedo, and land cover. To know the vegetation health during pre-monsoon and monsoon, we used MOD13C2 version-6 monthly Enhanced Vegetation Index (EVI) at the 0.05° Climate Modelling Grid (CMG) resolution and MCD15A2H version-6 8-day composite Leaf Area Index (LAI) product with 500 m pixel size. These datasets are averaged over 20 years from 2001 to 2020 for pre-monsoon (April–May) and monsoon (June-September) seasons. The land use and land cover data of 2010 is obtained from MODIS Terra+Aqua combined land cover product (MCD12Q1) based on International Geosphere-Biosphere Programme (IGBP) classification.

We used the daily maximum temperature from the IMD[35] for 1981–2020, and monthly EVI from 2001–2020 masked over Indo-Gangetic Plain to obtain the climatology of maximum temperature and EVI. Apart from this, we obtained the combined Dark Target and Deep Blue AOD at 550 nm for land and ocean from the level-3 MODIS gridded atmosphere daily global joint product (MOD08D3) at a spatial resolution of 1° CMG resolution and daily Land Surface Temperature (LST) at 0.05° CMG resolution from MOD11C1 version-6 product to analyse the relationship between AOD and daytime LST through a Granger Causality Test.

**Granger causality test**. The Granger causality test, first used in econometrics, is a statistical test that attempts to find a causal relationship between two time series[51]. A variable *x* Granger-cause another variable *y* when the past values of both *x* and *y* can better predict future values of *y* as compared to using the past values of *y* alone. Here, we employ the Granger causality test in the vector auto-regression model (VAR) framework for finding a causal relationship between two pairs (ET~LST and

AOD~LST) of 8-day composite datasets of April–May from 2001 to 2020 over UP and Bihar. We intend to check the role of irrigation through ET datasets; hence, daytime LST is used to establish a statistical relationship between ET and temperature. Here, the null hypothesis for the test is that lagged ET, or AOD values, do not explain the variations in daytime LST.

**Backward air-parcel trajectories**. To understand the source of air moisture over the two largest states of the Indo-Gangetic Plain, we calculate backward air-parcel trajectories for seasonal (April–May) maximum wet-bulb temperature days during pre-monsoon over UP and Bihar for 2001–2020 using Hybrid Single-Particle Lagrangian Integrated Trajectory (HYSPLIT) model from the NOAA's Air Resources Laboratory (https://www.ready.noaa.gov/hypub-bin/trajtype.pl?runtype=archive)[52]. The wet-bulb temperature[53] is calculated from near-surface hourly pressure, specific humidity, and temperature datasets[54]. Two sources were chosen; one in Bihar (26°N 86°E) and another in Uttar Pradesh (27°N 80.5°E), for ending air trajectory at 9:00 UTC (local time 14:30) at 50 m above ground level. It is assumed that maximum wet-bulb temperature occurs at 14:30 local time in the afternoon based on a study on extreme wet-bulb temperature in the South-Asian region[55].

## Data availability

All data that have supports the findings of this study are available upon request, including the raw WRF-CLM output. The post processed WRF-CLM output is publicly available in the Zenodo repository at https://doi.org/10.5281/zenodo.6064524. The final agricultural census-based pre-monsoon irrigation data are available at https://doi.org/10.5281/zenodo.6064524 and monsoon irrigation data are available at https://doi.org/10.25584/data.2019-08.903/1548406. The final irrigation data are prepared using agricultural census at district level containing area under the crop, crop irrigated area and sources of irrigation available at https://aps.dac.gov.in/LUS/Public/Reports.aspx and crop production statistics available at https://aps.dac.gov.in/APY/Public_Report1.aspx. The irrigation withdrawal data from Huang et al. is available at https://doi.org/10.5281/zenodo.1209296 and FAO area equipped for irrigation is available at https://www.fao.org/aquastat/en/geospatial-information/global-maps-irrigated-areas/latest-version/. The MODIS data for different variables: ET, EVI, LAI, LULC and LST are available at https://lpdaac.usgs.gov/products/. The MODIS AOD data is available at https://doi.org/10.5067/MODIS/MOD08_D3.061. The IMD temperature data is available at https://www.imdpune.gov.in/Clim_Pred_LRF_New/Grided_Data_Download.html. The ERA5 reanalysis dataset is available at https://doi.org/10.24381/cds.adbb2d47 and ERA-I reanalysis dataset for WRF-CLM is available from https://rda.ucar.edu/datasets/ds627.0/. Source data are provided with this paper.

## Code availability

All the codes that contributes to the irrigation data preparation, simulation output analysis and statistical analysis will be provided by the corresponding author upon reasonable request.

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

## Acknowledgements

This work is financially supported by the Department of Science & Technology (SPLICE—Climate Change Programme), India, through Project No. DST/CCP/CoE/140/2018 and Department of Science and Technology (Swarnajayanti Fellowship Scheme), India, through Project No. DST/SJF/E&ASA-01/2018-19;SB/SJF/2019-20/11. The lead author acknowledges the Fellowship support through DAAD In-Region Scholarship Programme South Asia.

## Author contributions

R.J., A.M. and S.G. conceived the work and developed the concept of the study. R.J. and A.D. prepared the irrigation datasets. A.D. incorporated representations of census-based irrigation application, groundwater pumping, and paddy field in WRF-CLM4. R.J. performed the numerical modelling, completed the data analysis, and wrote the manuscript with inputs from A.M., S.G. and R.M.K. All authors edited and revised the manuscript.

## Competing interests

The authors declare no competing interests.
