## [Peer Review File · Nature Communications]

Limited influence of irrigation on pre-monsoon heat stress in the Indo-Gangetic PlainReviewers' Comments:

Reviewer #1:

Remarks to the Author:

Please see the attachment.

Highlights:

The main result from the study is that the agricultural irrigation does not play a significant role in exacerbating heat stress over Gangetic Plains as has been reported in recent studies, especially in their ref. 4 (Mishra et al., 2020 Nature Geoscience). As a matter of fact, it appears that this manuscript is an extension of the previous work (ref. 4) that tried to estimate the influence of irrigation of heat extremes.

Comments/Questions/Suggestions:

There are multiple issues that come to my mind when I read the paper:

1. The authors claim that heat stress is most prominent in the months of **April and May**, which is not entirely true. This is apparent from their reference 19 study Figure 6 and Table 1a&b, which shows that heat stress is prevalent in June as well, and to some extent in July. April and July, however, show very limited heat stress; only **May and June** are the main heat stress months. Even the Extended Data Figure 1 also supports June as an important month for extreme temperatures. So, the period considered for the study is needed to confirm again. I wonder why the authors didn't consider June as the heatwave season. The other study authors have cited (ref. 20) to support the claim that April-May is the main heat stress season is only for one specific heat event (taken as May and June of 2015). This study does not consider April.
2. The authors note that April-May as "minimal irrigation" period. How can we then find a robust influence of "irrigation" on heat stress? Also, it is clear from Extended Figure 1 that both April and May have limited irrigation,

while significant irrigation takes place in June. It is only the month of June that has both high temperatures and high irrigation— a sole potential candidate month for robustly estimating the influence of irrigation on heat stress. Considering April-May is only expected to 2% increase in humidity due to irrigation, since there is minimal irrigation.

3. The paper doesn't finally discuss/quantify the impact of irrigation on "heat stress", which is the title. I suggest heat stress be defined first (as is done in previous studies), and then we can try to estimate the role of irrigation on heat stress.
4. Figure 2 shows that in the AGR simulations, the land surface temperature (LST) is only 0.2°C below as compared to no-agriculture scenario in UP and Bihar parts of the Gangetic Plains. This is highlighted to state that irrigation has limited influence of heat stress. The authors fail to notice/discuss the other major part of the Gangetic plains, which shows significant declines of LST (1.5°C below), as reported by ref. 4. Same argument can be dropped for Figure 3d, where wet-bulb temperature (T_w) shows contrasting changes in different parts of the Gangetic Plains.
5. Figure 4 and the text on moisture source do not seem to directly relevant to the title of the manuscript. In Figure 4 (a,b), the authors aim to show that concentration of aerosols is negatively correlated with temperature. This has not been related to irrigation-LST relationship. An important observation that can be made from this figure when compared with Figure 3 is that: we do not see a significant spatial contrast in AOD and MODIS-LST over the Gangetic Plains that we can be noticed in Figure 3. I invite the authors to understand the implications from this observation.

Other Comments

L9: The references cited in the main text suggest that May and June as the main season of heat stress.

L29: In the abstract, major cropping seasons are mentioned as June to Sept and Nov. to Feb. Are there crops failing in non-cropping season?

L32 to L33: Please rephrase. These studies don't specially discuss the role of human water management practices on heat stress.

L44-45: Again, the argument that the previous studies, 4,5, and 6, use annual irrigation is not correct. For example, reference 6 clearly mentions that irrigation is "confined to the crop growing season". Also, see Figure 2, Figures S1 and S2 (and others) of reference 5, where the author specially focus on showing the changes of extreme heat indices during summer season.

L70-72: Here the authors fail to acknowledge that highest temperatures happen in June also, when agricultural activities are significant.

I believe the section "Temperature extremes and non-cropping season" is not very helpful and may be removed. In this section, the authors intend to mention that temperature extremes are likely to happen in April and May (Non-cropping season). But when tied to Extended Data Fig 1, there is significant agricultural activity during June and yet has higher temperature than April. So, I think the point that hot temperatures only happen during dry seasons is

not true. As a matter of fact, the early cited paper ref. 19, shows small number of heatwaves during April and large number during June.

In Figure 4: Only first two panels are useful. Other can be derived from the first two panels

Panels a and b show that high aerosol depth implies low land surface temperature. Multiple questions arise.

1. Can we quantify that how much "heat stress extremes" have decreased?
2. Can we explain the spatial inhomogeneity of the decrease in temperature in Figure 3, where UP and Bihar shows little decrease while northwest portion shows large decreases.
3. Can we extrapolate daily relationships to heat stresses?

Methods needs to be improved for clarity. Here are some suggestions that authors can implement.

What variable does the census data include?

Is the April May (summer) crop production used anywhere in the study?

What irrigation-related variables are used in UP and Bihar states and in other states?

Reviewer #2:

Remarks to the Author:

This study follows on and attempts to rebut that of Mishra et al 2020 on the role of irrigation in India in driving increases in moist heat stress. This is an important topic worthy of further investigation and no study should be considered the last word on the topic. From that point of view I find the study to be relevant and worthy of publication. The main claim of the study is that, in the premonsoon season, (Apr-May) irrigation is not much practiced, so that any trend that is ascribed to irrigation, or modeling that supports it, must be wrong. This is certainly an idea worth following up on, since Mishra et al did not provide observational justification for the specific irrigation numbers that they employed in their modeling. But, this also means to make the point convincingly, the irrigation data themselves become a major bone of contention and should be extensively documented and supported.

One problem with this study is that the main results of the study hinge completely on an irrigation data set that is poorly described, buried in supplemental material, and unverified. There are several other problems which I will describe more below and I will also suggest some solutions to these problems. In the end, I see value in this study, but I think extensive rethinking, rewriting, and some new experiments and recalculation are in order.

First, let me start with what I think we all agree on. Moist heat stress in the region probably arises from a complex interplay of local and regional hydrology including moisture transport and irrigation, as well as inputs of radiation, all of which are impacted by atmospheric circulation, rainfall, clouds and aerosols. Indeed, this study agrees in important ways with the Mishra et al study. Their MOD experiment, in which water is added as need to maintain saturation produces results that are very similar to those in the Mishra study. So, if enough water is added, I think that both studies reach the same answer. Major cooling, increases in humidity, reductions in boundary layer height. Indeed the MOD simulations in this study look surprisingly similar to those in Mishra et al. The main role of WRF/CLM simulations in the Mishra et al paper was to as a sensitivity study to demonstrate the physical mechanisms by which irrigation might cause enhanced moist heat stress, and in that sense this study does not change that picture. While not discussed much in the Mishra et al study, a series of sensitivity tests to varying irrigation were actually carried out and discussed in supplemental material (Figure S11 and S13) and the conclusions for those sensitivity tests were that there was a simple linear relationship between the degree of irrigation and the impact on relevant parameters.

Thus, the only really important issue that is addressed in this new paper is whether the AGR irrigation scenario is so much different, and so much closer to reality, that it somehow invalidates the argument that part of the trend in wet bulb temperature ascribed in the Mishra et al study was due to irrigation. It is clear from Figure 1b that (in their estimation) irrigation in UP and Bihar is quite close to zero in the pre-Monsoon season. So it is actually not surprising that their results resemble the weaker irrigation states studied in Mishra et al (Figure S13). The justification given in the text for this estimation is, frankly, unclear (245-275). Especially how the irrigation is distributed specifically during the Apr-May window (there is no description of how that is done and the source website (<https://aps.dac.gov.in/LUS/Public/Reports.aspx>) does not have that information in any clear way. I'm not saying this was done improperly, but given that the entire paper depends on it, I can not support publication until this element is transparent and validated.

I further raise this concern because of this 269-271: "The average irrigation water withdrawal data from Huang et al.²⁹ and the irrigated area fractions from Food and Agricultural Organisation (FAO) Global Map of Irrigated Areas (GMIA)²⁷ are used over the model domain where agricultural census estimates are not available." The authors have used a completely different dataset everywhere else in India, and it is very clear in the regions just North and West of UP, that the Huang et al dataset has strong irrigation. There is a noticeable, and unphysical break in all the AGR results across this line between the author's derived water use (UP&Bihar) and the rest of the map (this shows up in Figure 2b, and 2C suggesting that MOD is not actually in grave disagreement with the Huang et al dataset, which is partially used in AGR). This shows up through out Figure 3 as well (and the rest of the paper).

Huang et al, from an independent study, which is used outside of 2 states in this study, appears to be not far off from MOD, but completely different than the irrigation results the authors have derived (from opaque methods) in UP and Bihar. To improve this paper, a set of WRF simulations with Huang et al irrigation needs to be performed for comparison and a separate section comparing the authors' estimates in UP and Bihar against those in Huang et al (and presumably a statement why theirs are better).

In addition to that, which is my main scientific concern, there are several technical issues that should be addressed, and these will require further simulations.

First, it is not just irrigation that is the issue, it is the soil moisture. These simulations are begun and spun-up for a relatively brief time. What is the initial soil moisture distribution? In the real world, irrigation signals may persist within soil moisture for months anomalies. Unless the soil moisture is initialized as saturated and then spun up, then, in the absence of rain and (weak specified) irrigation then the soil moisture will be much drier in the AGR simulations than is perhaps realistic. A sensitivity tests to this (by beginning with wet soils) would help.

Second, as far as I can tell wet bulb is being calculated from averaged quantities, not for the original hourly (or 3-hourly) output. Since this is a highly non-linear quantity that varies strongly over the diurnal cycle, daily or even 12 hourly means of T, RH, etc are not sufficient. Wet bulb should be calculated at high resolution (as it was in the Mishra et al study). 1-2° Tw errors can arise from this and the offset may be strongly impacted by exactly the boundary layer process that are crucial for this study.

Third, and in a similar vein, the Stull approximation to wet bulb temperature is substantially inaccurate away from typical midlatitude conditions and should not be used in this study. Many more accurate techniques exist for this computation and freely available packages can be used to do it. This should be done.

Reviewer #3:

Remarks to the Author:

Anomalous review for "Limited influence of irrigation on heat stress in the Gangetic Plain"

This study explores the over-estimation of irrigation impacts on heat stress over the Indian regions. They use the WRF model combined with the satellite dataset to demonstrate previous studies might significantly over-estimate (overemphasize) the irrigation's cooling impacts. They further argue that aerosol might play a more prominent role in surface cooling than surface irrigation. The authors raise one critical point of the seasonality issue: the irrigation season and dry season do not coincide. The previous studies may misinterpret; further, the model's irrigation scheme may over-estimate the irrigation water amount, thus, over-estimating the cooling effect from irrigation.

In general, this study provides some new information to the community about how we can better represent the irrigation impact in the model and consider both temporal impacts of irrigation and the local mean climate (whether the irrigation season is the same as the dry season).

The results look interesting, and this study provides some valuable information for the community. However, this paper needs to include more discussions on the uncertainties and scientific questions exploration, and some comments need to be addressed. These are outlined below.

1. Besides showing the annual, pre- and post-monsoon irrigation fraction, how about the irrigation water? During the monsoon season, the irrigation amount should be minimal as well due to a large amount of rainfall? Or it's the other way around? So, only showing the irrigation fraction is not

enough; please also demonstrate and discuss the irrigation water amount's temporal variations.

2. The authors have also argued that limited role in hot extremes during fewer irrigation activities in the pre-monsoon season. More analysis on the irrigation activities is needed to be more convincing.

3. Figure 4c is interesting, and can the authors also show the scatter plot for the whole India region?

4. Validation: Extended data figure 4 may not be enough for the model-observation validation. How about showing the pattern correlation?

5. Does the WRF include both aerosol direct and indirect effects? Using WRF to simulate the aerosol's effect on such surface cooling tendency will be a nice add-on. Then, the authors can further discuss the aerosol's direct and indirect effect on the surface cooling here.

6. WRF provides various PBL and convection parameterizations, so will the different parameterizations affect the results shown here? Some sensitivity tests for this regard can make the results more robust.

7. Using the reanalysis data as the boundary condition to force the WRF model may lack the remote impacts, such as the whole monsoon strength? Some studies already mentioned that the critical role of India's massive irrigation may affect India's monsoon strength. Will the weak signal in this study come from the lack of remote impacts or large-scale response from Indian irrigation? Or, the further question will be: whether the WRF platform's boundary condition affects the results in this study?

8. The authors also argue that "However, the significant irrigation expansion is observed during monsoon and high aerosol loading measured during dry pre-monsoon, which depicts that cooling trend during pre-monsoon is associated strongly with high aerosol loading than limited pre-monsoon irrigation" based on Shukla, S. P., Puma, M. J. & Cook, B. I. The response of the South Asian Summer Monsoon circulation to intensified irrigation in global climate model simulations. *Clim. Dyn.* 42, 21–36 (2014). Not sure Shukla et al., 2014 can show the significant irrigation expansion? What does it mean?

Limited influence of irrigation on pre-monsoon heat stress in the Indo-Gangetic Plain

Manuscript No: NCOMMS-21-32895A

Response to Reviewers

We thank the editorial board and the reviewers for their insightful and encouraging comments. Below we provide a point by-point response to the reviewers' comments and details of the actions taken. Figures, tables, line and page numbers mentioned in this document refer to the revised manuscript, unless specified otherwise. Reviewer comments are shown in **bold**. Author responses are shown in plain text.

1 Reviewer#1

1.1 Comment 1

The main result from the study is that the agricultural irrigation does not play a significant role in exacerbating heat stress over Gangetic Plains as has been reported in recent studies, especially in their ref. 4 (Mishra et al., 2020 Nature Geoscience). As a matter of fact, it appears that this manuscript is an extension of the previous work (ref. 4) that tried to estimate the influence of irrigation on heat extremes.

Response

We thank the reviewer for understanding the main objective of this study, as mentioned in the first sentence. However, we beg to differ that this manuscript is an extension of previous work. The recent earlier study (Mishra et al.¹) used a regional land-atmosphere model with pre-monsoon (April and May) irrigation and showed significant impacts of irrigation feedback on the moist heat stress. We showed that such a conclusion resulted from a model artifact that does not consider the regional crop calendar. The pre-monsoon season is not a major cropping season in India with minimal irrigation applications. With the Governmental agricultural census-based irrigation data and regional land-atmosphere model, designed for Indian agricultural practices, we showed that irrigation have limited effect on the moist heat stress over the Indo-Gangetic basin in the pre-monsoon season. We also would like to highlight a key message from our work: any regional representation of human-natural climate system needs consideration of regional characteristics and processes; the use of models adopted from different regions may lead to erroneous conclusions.

1.2 Comment 2

The authors claim that heat stress is most prominent in the months of April and May, which is not entirely true. This is apparent from their reference 19 study Figure 6 and Table 1a&b, which shows that heat stress is prevalent in June as well, and to some extent in July. April and July, however, show very limited heat stress; only May and June are the main heat stress months. Even the Extended Data Figure 1 also supports June as an important month for extreme temperatures. So, the period considered for the study is needed to confirm again. wonder why the authors didn't consider June as the heatwave season. The other study authors have cited (ref. 20) to support the claim that April-May is the main heat stress season is only for one specific heat event (taken as May and June of 2015). This study does not consider April.

Response

The present study analyzed the feedback from Irrigation to the pre-monsoon heatstress. In India, the pre-monsoon season refers to April and May, followed by the Monsoon season with the climatological onset date as 1st June. We agree with the reviewer that in the early monsoon season (June), specifically in the late monsoon onset years, the heat stress is high, with high irrigation feedback from the monsoon (Kharif season) irrigation practices. Further, the major cropping season known as the Kharif season in India coinciding with the Indian summer monsoon, have higher area under the crop and crop irrigated area than the pre-monsoon season (April and May) as shown in Figure 1b,c. We expect the influence of irrigation on hot extremes during the pre-monsoon season to be different from that during the monsoon season. Thus, we studied the pre-monsoon season (April-May) and have now added “pre-monsoon” in title of the manuscript.

In addition, we have updated the references (19,20) with relevant references that have studied hot extremes during April-June and clarified as *"India experiences hot extremes resulting in human mortality and agricultural crop failures during the pre-monsoon season (April-May) and early monsoon season, in June²⁻⁴".* (Line 29-30)

Further, we modified some sentences *"In India, large-scale irrigation is observed only during the monsoon and post-monsoon seasons: Kharif and Rabi, extending from June to September and November to February, respectively. The rest of the months, March to May is usually hot and dry without extensive agricultural activities because of cropping pattern^{5,6} and government policies related to groundwater conservation⁷. Heat stress associated with hot extremes are observed during the pre-monsoon season (April and May) and in the early monsoon season (June), specifically in the late monsoon onset years with high irrigation feedback from the monsoon (Kharif season) irrigation practices. Recent studies^{1,8} on irrigation feedbacks during pre-monsoon season used land surface models to estimate irrigation amounts in the absence of region-specific irrigation data over the Indian region. Further, these studies used annual irrigated areas instead of pre-monsoon seasonal area fraction, failing to account that agricultural activities and irrigation in the field are minimal during pre-monsoon hot extremes. While such approaches may be suitable for other regions globally, they may overestimate pre-monsoon irrigation amounts in the Indo-Gangetic Plain".* (Lines 44-57)

1.3 Comment 3

The authors note that April-May as "minimal irrigation" period. How can we then find a robust influence of "irrigation" on heat stress? Also, it is clear from Extended Figure 1 that both April and May have limited irrigation, while significant irrigation takes place in June. It is only the month of June that has both high temperatures and high irrigation— a sole potential candidate month for robustly estimating the influence of irrigation on heat stress. Considering April-May is only expected to 2% increase in humidity due to irrigation, since there is minimal Irrigation.

Response

We echo with the reviewer that the feedback from irrigation is minimal during the pre-monsoon season and highest during the early monsoon season. Recent studies^{1,8-10} showed that the irrigation has significant feedback to the pre-monsoon (April and May) heatwaves. Our objective is to highlight the same and demonstrate minimal feedback from irrigation to pre-monsoon heat stress using region-specific data and a region-specific model. Here we showed that the influence of irrigation to the pre-monsoon heat stress is not robust and stems from model artifact. Further, to add robustness to our rebuttal, we have performed three experiments with different parameterization schemes as described in Supplementary Note 3. We argue here that any regional representation of human-natural climate system needs consideration of regional characteristics and processes; the use of models adopted from different regions may lead to erroneous conclusions.

We have now modified and added some sentences to show the relevance of studying the pre-monsoon season, "*Further, the dip in temperature with an increase in EVI in June depicts the onset of the Indian summer monsoon along and the beginning of the major cropping season (Kharif season). Monsoon season/Kharif season observes extensive irrigation compared to pre-monsoon (April and May) suggesting the contrasting irrigation feedback on hot extremes during two seasons. Thus, it can be understood that high temperature favouring the hot extremes in pre-monsoon coincides with minimum agricultural activities and limited irrigation, as cropland is the major land-use land cover class over the Indo-Gangetic Plain (Supplementary Fig. 2), need to be studied using region-specific data and a region-specific model*". (Lines 78-86)

1.4 Comment 4

The paper doesn't finally discuss/quantify the impact of irrigation on "heat stress", which is the title. I suggest heat stress be defined first (as is done in previous studies), and then we can try to estimate the role of irrigation on heat stress.

Response

We agree with the reviewer that heat stress should be defined first. We have adopted the approach by Mishra et al.¹ to define dry heat stress and moist heat stress using dry-bulb temperature and wet-bulb temperature, respectively. We have now modified and added some phrases in the introduction section.

"However, non-irrigated areas under dry conditions favour the hot extremes through a rise in sensible heat and reduced evapotranspiration, causing dry heat stress on human body^{11,12}."

(Line 25-27)

"Hot extremes under humid conditions can reduce the human body capacity to maintain a healthy body temperature by perspiration, thereby inducing moist heat stress. Therefore, it is imperative to adequately quantify the irrigation feedback on dry heat stress and moist heat stress given by dry-bulb temperature and wet-bulb temperature, respectively, in this region."

(Lines 37-40)

1.5 Comment 5

Figure 2 shows that in the AGR simulations, the land surface temperature (LST) is only 0.2° below as compared to no agriculture scenario in UP and Bihar parts of the Gangetic Plains. This is highlighted to state that irrigation has limited influence of heat stress. The authors fail to notice/discuss the other major part of the Gangetic plains, which shows significant declines of LST (1.5°C below), as reported by ref. 4. Same argument can be dropped for Figure 3d, where wet-bulb temperature (Tw) shows contrasting changes in different parts of the Gangetic Plains.

Response

We thank the reviewer for highlighting the contrasting changes in the different parts of the Indo-Gangetic Plain. Here, we structure our response in several bullet points.

- The contrasting response over UP and Bihar and other parts of Gangetic Plain (Punjab and Haryana) is due to the different irrigation data (census-based irrigation data in UP-

Bihar and Huang et al.¹³ irrigation data in other parts) used in the respective region. We understand this is not the best way to do it; however, we focused on the UP, and Bihar states as only these states provide the measured irrigation data. For other regions, the data was taken from that derived by Huang et al.¹³ The irrigation data for other regions does not present the actual estimated data.

- Now, we have updated our region of interest to the Indo-Gangetic Plain (UP, Bihar, Haryana, Punjab and Rajasthan) based on updated crop calendar (Supplementary Table 1) and crop production data from Indian Agricultural statistics⁵ for the pre-monsoon/summer season and prepared the required irrigation datasets as described in Supplementary Note 1.
- With the updated irrigation dataset, the LST and wet-bulb temperature and other meteorological variables response to irrigation during April-May are consistent over Indo-Gangetic Plain except over a sugarcane-growing areas where higher cooling and moistening is observed since sugarcane is a water-intensive crop. Now, the Figure 2 and Figure 3 shows consistent irrigation feedback on different variables over the different parts of Indo-Gangetic Plain.
- We also like to mention that the irrigation withdrawal data from Huang et al.¹³ over Indo-Gangetic Plain overestimates the irrigation amount and hence higher evaporative cooling and moistening as shown in added Supplementary Figure 9. The most probable reason is the tendency of water models to estimate higher irrigation amounts for dry soil conditions during pre-monsoon season. However, the agricultural statistics data shows that the region practices limited irrigation during pre-monsoon season.

1.6 Comment 6

Figure 4 and the text on moisture source do not seem to directly relevant to the title of the manuscript. In Figure 4 (a,b), the authors aim to show that concentration of aerosols is negatively correlated with temperature. This has not been related to irrigation-LST relationship. An important observation that can be made from this figure when compared with Figure 3 is that: we do not see a significant spatial contrast in AOD and MODIS-

LST over the Gangetic Plains that we can be noticed in Figure 3. I invite the authors to understand the implications from this observation.

Response

The response to the comments is structured in the points below.

- Firstly, there is limited irrigated area and irrigation water use over the Indo-Gangetic plain during the pre-monsoon season, and it does not have a large influence on near-surface temperatures. Hence, there must be other factors that must influence the temperature and humidity in the region. Here, Figure 4 relates LST cooling with aerosol, and the analysis of moisture source reveals the presence of non-local moisture in the region as shown in Supplementary Figure 19. The following are modified for showing the relevance of aerosol and non-local soil moisture, respectively.

“Both irrigation activities and aerosol loading have increased over Indo-Gangetic Plain since 1980 or before. However, a significant irrigation expansion is observed only during monsoon¹⁴, and aerosol loading increasing since 1980s is high during pre-monsoon¹⁵, which depicts that the cooling and moistening trend that is observed during pre-monsoon is associated with other factors like high aerosol loading¹⁶ than limited pre-monsoon irrigation over Gangetic Plain (Uttar Pradesh and Bihar).” (Lines 248-253)

“The moist air coming from these sources during the extreme moist heat condition can be the additional reason for increased moist heat stress over Gangetic Plain.” (Line 258-260)

- In addition, Granger's Causality test based on observational data shows ET (proxy of irrigation amount) do not granger causes LST over UP and Bihar. However, AOD granger causes LST, which makes our statement stronger that there is a limited influence of irrigation on heat stress due to irrigation compared to aerosols over UP and Bihar. However, both the ET and AOD granger causes LST over the Punjab and Haryana, which depicts that irrigation influences temperature in this region.
- The significant spatial contrast in Figure 3 is because of two irrigation data sources (census-based irrigation data in UP-Bihar and Huang et al. irrigation data in other parts) as described in response to comment 5. Now, we have updated the region of interest

with census-based irrigation data over the Indo-Gangetic Plain to as described in Supplementary Note 1. Now, the difference between AGR and CTL experiment showing census-based irrigation feedback on different variables (shown in Figure 3) do not have spatial contrast across the Indo-Gangetic Plain like remotely sensed MODIS AOD and LST.

1.7 Comment 7

L9: The references cited in the main text suggest that May and June as the main season of heat stress.

Response

The present study analyzed the feedback from irrigation to the pre-monsoon (April and May) heatstress. We agree with the reviewer that in the early monsoon season (June), specifically in the late monsoon onset years, the heat stress is high, with high irrigation feedback from the monsoon (Kharif season) irrigation practices. In addition, the major cropping season (Kharif season) in India coinciding with the Indian summer monsoon, have higher area under the crop and crop irrigated area than the pre-monsoon season (April and May) as shown in Figure 1b,c. We expect the irrigation feedback on heat stresses during the pre-monsoon season to be different from that during the monsoon season. Thus, we studied the pre-monsoon season (April-May). We have now cited relevant references in the main text that have studied hot extremes during April and May.

“India experiences hot extremes resulting in human mortality and agricultural crop failures during the pre-monsoon season (April-May) and early monsoon season, in June²⁻⁴.” (Line 29-30)

1.8 Comment 8

L29: In the abstract, major cropping seasons are mentioned as June to Sept and Nov. to Feb. Are there crops failing in the non-cropping season?

Response

Major water-intensive crops like paddy and wheat are grown during monsoon (Jun-Sep) and post-monsoon season (Nov-Feb), also known as Kharif and Rabi season respectively in Indo-Gangetic Plain. In the dry summer season, paddy farming is only limited to small areas in the east Indo-Gangetic Plain (Uttar Pradesh and Bihar). Other crops like maize, sunflower, and

grams are also grown in limited areas in the pre-monsoon season. In some parts of UP, Punjab, and Haryana, sugarcane is sown in Feb-March and harvested in Oct-Nov.

These information are also reported in FAO irrigated crop calendar database⁶ for India at monthly scale (<https://www.fao.org/aquastat/en/databases/crop-calendar/>), where April-May is shown to have zero or least percentage of crop irrigated area for different crops. In summary, agriculture is limited in the hot and dry pre-monsoon season due to cropping pattern⁵ and government policies⁷ that regulate the timing of paddy crops to reduce groundwater depletion, especially in Punjab and Haryana.

Now, we have modified introduction section and cited the relevant document *“In India, large-scale irrigation is observed only during the monsoon and post-monsoon seasons: Kharif and Rabi, extending from June to September and November to February, respectively. The rest of the months, March to May is usually hot and dry without extensive agricultural activities because of cropping pattern^{5,6} and government policies⁷ related to groundwater conservation”*. (Lines 44-48)

In addition the section “Pre-monsoon temperature extremes and non-cropping season” shows the difference between cropping activities pre-monsoon and post-monsoon season based on agricultural census data and satellite data.

1.9 Comment 9

L32 to L33: Please rephrase. These studies don't specially discuss the role of human water management practices on heat stress.

Response

Here, the sources of irrigation are surface water and/or groundwater. Therefore, we have taken irrigation activities as human water management practices. We have clarified this in the revised manuscript. The L32 to L33 is rephrased as

“Other studies^{1,2,17} also discuss the confounding role of human water management practices such as irrigation on hot extremes over Indo-Gangetic Plain.” (Line 32-34)

Here are the relevant excerpts from the referenced studies

Mishra et al.¹ shows the influence of irrigation on heat stresses in their main results as *“Here we analysed a combination of in situ and satellite-based datasets and conducted meteorological model simulations to show that irrigation modulates extreme moist heat.”*

Joshi et al.² describe irrigation's role on heat extremes as *"Due to intensive irrigation over the Indo-Gangetic plains, the vegetation and evapotranspiration have increased substantially. This increase in evapotranspiration has led to the smaller portion of sensible heat flux (Fig. 4d) versus latent heat (Fig. 4e) over most parts of the Indo-Gangetic plains during the global warming period as compared to non-global (for details see Fig. S4), which has resulted in a decrease in near-surface air temperatures over that region."*

Oldenborgh et al.¹⁸ describe irrigation's role on heat extremes as *"Due to the increase in humidity from irrigation and higher sea surface temperatures (SSTs), these indices have increased over the last decades even when extreme temperatures have not."*

Therefore, these studies indeed discuss the role of human water management practices on heat stress.

1.10 Comment 10

L44-45: Again, the argument that the previous studies, 4,5, and 6, use annual irrigation is not correct. For example, reference 6 clearly mentions that irrigation is "confined to the crop growing season". Also, see Figure 2, Figures S1 and S2 (and others) of reference 5, where the author specially focus on showing the changes of extreme heat indices during summer season.

Response

We thank the reviewer for highlighting this important point. The responses are structured in bullet points for each referenced study.

- Ref. 4 (Mishra et al.¹) focused on April-May and prescribed the irrigation area and irrigation water as quoted below from their method section.

"The fractional area under irrigation from the Food and Agriculture Organization irrigation map was obtained to implement the irrigation scheme in the Noah land surface model coupled with WRF. Irrigation was applied as precipitation when the root-zone soil moisture fell below the field capacity"

Here, the authors of ref. 4 have used the FAO irrigation map that represents the annual irrigation area fraction which is higher than the irrigation fraction observed during April-May. The FAO annual irrigation fraction is shown in Figure 1a of Mishra et al.¹. Further, irrigation was applied as deficit water calculated as the difference between root-zone soil moisture at any point and soil moisture at field capacity.

- Ref. 5 and 6 (Thiery et al.^{8,9}) have adopted a similar irrigation scheme, where irrigation fraction area is taken from Siebert et al.¹⁹, which is the same as FAO data. The data shows the area equipped for irrigation around 2005 in the percentage of the total area on a raster. (<https://www.fao.org/aquastat/en/geospatial-information/global-maps-irrigated-areas/latest-version/>)

Next, the irrigation water is calculated as a deficit between target and actual soil moisture. The target soil moisture is calculated using Equation 1 given in ref. 5.

$$w_{\text{target},i} = (1 - 0.7) * w_{o,i} + 0.7 * w_{\text{sat},i}$$

Here, the value 0.7 is chosen empirically to match global, annual irrigation amounts in CLM4 with values observed around 2000. However, the regions like Indo-Gangetic Plain where the soil moisture is very low during the pre-monsoon months, the model tends to estimate a higher amount of deficit water over the larger annual irrigation area fraction, thereby over-estimating the irrigation amount over the region.

- Ref. 6 mentions "*more water is applied during hot days (Supplementary Fig. 8) as their occurrence typically coincides with crop growing seasons and in many regions also with precipitation deficit*". While this may be true for other regions of the world, Indo-Gangetic Plain has less irrigation area fraction during hot days of pre-monsoon season.
- Figure 2 of ref 5 (reproduced below) compares the annual irrigation rates over SREX regions from observation and simulation. However, as stated by the reviewer, the author especially focuses on showing the changes of extreme heat indices during the hot days which is during pre-monsoon season in Indo-Gangetic Plain. Hence, a comparison of irrigation rates of pre-monsoon season between model simulation and observation would provide a clear picture.

Figure 2. Annual irrigation rates Q_{irr} (mm yr^{-1}) averaged over the SREX regions as observed around 2000 [Helkowski, 2004; Sacks et al., 2009] and simulated by CESM from 1981–2010.

To summarize, these studies have not incorporated seasonality in the irrigation prescription, which is crucial for regions like Indo-Gangetic Plain. The following sentences have been slightly modified for clarity (Lines 51-57).

"Recent studies^{1,8} on irrigation feedbacks during pre-monsoon season used land surface models to estimate irrigation amounts in the absence of region-specific irrigation data over the Indian region. Further, these studies used annual irrigated areas instead of seasonal area fraction, failing to account that agricultural activities and irrigation in the field are minimal during pre-monsoon hot extremes. While such approaches may be suitable for other regions globally, they may overestimate pre-monsoon irrigation amounts in the Indo-Gangetic Plain."

1.11 Comment 11

L70-72: Here the authors fail to acknowledge that highest temperatures happen in June also, when agricultural activities are significant.

Response

We agree with the reviewer that June also witnesses high temperature and significant agricultural activities. However, as described in response to Comment 1 and Comment 2, we are interested in quantifying the influence of actual observed irrigation on heat stress during pre-monsoon months (April and May) which has not been explored yet.

Now, we have added and modified few sentences *"Heat stress associated with hot extremes are observed during the pre-monsoon season (April and May) and in the early monsoon season*

(June), specifically in the late monsoon onset years with high irrigation feedback from the monsoon (Kharif season) irrigation practices. Recent studies^{1,8} on irrigation feedbacks during pre-monsoon season used land surface models to estimate irrigation amounts in the absence of region-specific irrigation data over the Indian region. Further, these studies used annual irrigated areas instead of pre-monsoon seasonal area fraction, failing to account that agricultural activities and irrigation in the field are minimal during pre-monsoon hot extremes. While such approaches may be suitable for other regions globally, they may overestimate pre-monsoon irrigation amounts in the Indo-Gangetic Plain". (Lines 48-57)

1.12 Comment 12

I believe the section "Temperature extremes and non-cropping season" is not very helpful and may be removed. In this section, the authors intend to mention that temperature extremes are likely to happen in April and May (Non-cropping season). But when tied to Extended Data Fig 1, there is significant agricultural activity during June and yet has higher temperature than April. So, I think the point that hot temperatures only happen during dry seasons is not true. As a matter of fact, the early cited paper ref. 19, shows small number of heatwaves during April and large number during June.

Response

We thank the reviewer for the suggestion. However, the section "Temperature extremes and non-cropping season" aligns with our research question of investigating the irrigation feedback on heat stress during pre-monsoon season, which is a non-cropping season. In the early monsoon season (June), specifically in the late monsoon onset years, the temperature is high, with high irrigation feedback from the monsoon (Kharif season) irrigation practices. We expect the influence of irrigation on hot extremes during the pre-monsoon season to be different from that during the monsoon season. Thus, we study the pre-monsoon season "heat stress" and have now added "**pre-monsoon**" in title of the manuscript and subheading of result.

We now have modified some sentences *"The daily maximum temperature from Indian Meteorological Department (IMD) averaged from 1981-2020 over the Indo-Gangetic Plain, reaches the peak in May; however, the monthly enhanced vegetation index (EVI), an improved measure of vegetation density, averaged over the available period 2001-2020 shows minimum values during May (Supplementary Fig. 1). Further, the dip in temperature with an increase in EVI in June depicts the onset of the Indian summer monsoon along and the beginning of the major cropping season (Kharif season). Monsoon season/Kharif season observes extensive*

irrigation compared to pre-monsoon (April and May) suggesting the contrasting irrigation feedback on hot extremes during two seasons. Thus, it can be understood that high temperature favouring the hot extremes in pre-monsoon coincides with minimum agricultural activities and limited irrigation, as cropland is the major land-use land cover class over the Indo-Gangetic Plain (Supplementary Fig. 2), need to be studied using region-specific data and a region-specific model". (Lines 74-86)

1.13 Comment 13

In Figure 4: Only first two panels are useful. Other can be derived from the first two panels. Panels a and b show that high aerosol depth implies low land surface temperature. Multiple questions arise.

Response

We agree with the reviewer that Figure 4a,b shows the relation between high aerosol depth and low land surface temperature. The other statistical tests are performed to show land surface temperature and aerosol loading have statistically significant correlation and latter have causal effect on former.

1.14 Comment 14

1. Can we quantify that how much "heat stress extremes" have decreased?

Response

The quantification of the influence of aerosol loading on heat stress extremes requires WRF-Chem modeling, which is beyond the objectives of the present study. One recent study by Dey et al.¹⁶ has shown that high aerosol loading in India could mask the heat stress (quantified by the Wet Bulb Globe Temperature, WBGT) by 0.3-1.5°C in 2010. This recent finding has been cited in Line 59-62 and 228-230.

“Therefore, attribution of decreasing land surface temperature^{1,2} and rising wet-bulb temperature^{1,17,20} over the Indo-Gangetic Plain to irrigation¹ alone based on over-estimated irrigation overlook contribution from other factors like aerosol loading¹⁶ and remote moisture transport as important as irrigation.” (Lines 59-62)

“Model-driven studies have shown the direct and indirect effects of aerosol-radiation interaction on lowering the temperature²¹ and increasing the relative humidity¹⁶.” (Line 228-230)

1.15 Comment 15

Can we explain the spatial inhomogeneity of the decrease in temperature in Figure 3, where UP and Bihar shows little decrease while northwest portion shows large decreases.

Response

The contrasting response over UP and Bihar and other parts of Indo-Gangetic Plain is due to the different irrigation data (census-based irrigation data in UP-Bihar and Huang et al. irrigation data in other parts) used in the respective region. We understand this is not the best way to do it; however, we focused on the UP and Bihar states as only these states provide the measured irrigation data. For other regions, the data was taken from that derived by Huang et al.¹³ which does not present the actual estimated data.

Now, we have updated our region of interest to the Indo-Gangetic Plain (UP, Bihar, Haryana, Punjab, and Rajasthan) based on the updated crop calendar from Indian Agricultural statistics and crop production data for the pre-monsoon/summer season and prepared the required irrigation datasets as described in Supplementary Note 1. The updated simulation results do not show the spatial inhomogeneity of the decrease in temperature across the Indo-Gangetic Plain as both the UP and Bihar and Northwest portion (Punjab, Haryana) of Indo-Gangetic Plain are prescribed agricultural census-based irrigation data. Now, the Figure 2 and Figure 3 shows consistent irrigation feedback on different variables over the different parts of Indo-Gangetic Plain.

1.16 Comment 16

Can we extrapolate daily relationships to heat stresses? Methods needs to be improved for clarity. Here are some suggestions that authors can implement. What variable does the census data include?

Response

Here, we have simulated the variables at an hourly scale and averaged the hourly data to a daily scale. The maximum temperature is taken as a maximum of 24 hours. The dry heat stress and moist heat stress are represented by daily 2m-air temperature and wet-bulb temperature, similar to the definition adopted for model simulated output in Mishra et al.¹. Here, we are mainly interested in irrigation influence on these variables which are the indicator of heat stress. Further, we have shown the changes in seasonal 95th percentile of these variables due to

irrigation representing extreme heat stresses. These details are updated in the method section for clarity.

“We use IMD²² daily mean temperature to assess the model skill over the Indo-Gangetic Plain. The simulated meteorological variables (mean temperature, maximum temperature, minimum temperature, specific humidity, relative humidity, pressure, heat fluxes, and wet-bulb temperature) at an hourly scale is converted to daily scale and then averaged over the simulation period for each set of the experiment. The hourly wet-bulb temperature is calculated using hourly mean temperature, hourly dew-point temperature, and hourly surface pressure using the iterative procedure described by Stipanuk²³ available in NCL. The dry heat stress and moist heat stress is represented by daily 2m-air temperature and wet-bulb temperature similar to the definition adopted for model simulated output in Mishra et al.¹. The 95th percentile of daily maximum temperature (Tmax_95) and daily wet-bulb temperature (Tw_95) during the pre-monsoon season represent the extreme dry and moist heat conditions, respectively.” (Lines 320-331).

The census data includes **area under the crop, crop irrigated area, sources of irrigation and land use area**. Now, we have added the detailed description regarding the census-based irrigation data provided in Supplementary Note 1.

1.17 Comment 17

Is the April May (summer) crop production used anywhere in the study? What irrigation-related variables are used in UP and Bihar states and in other states?

Response

The crop production data for the summer season provides information on crops grown during the summer, hence indirectly referred in this study.

Now, we have updated the region of interest to the Indo-Gangetic Plain and updated the census-based irrigation data as described in Supplementary Note 1. The variables representing irrigation data used over the Indo-Gangetic plain (Rajasthan, UP, Bihar, Punjab and Haryana) are:

- Non-paddy irrigation use (mm/day)
- Paddy irrigation use (mm/day)
- Fraction of water from the ground (F_grd)
- Fraction of water from surface (F_Surf)

- Non-paddy irrigated area fraction
- Paddy irrigated area fraction

The other states are prescribed daily irrigation use (mm/day) provided by Huang et al¹³ based on Siebert et al.¹⁹ irrigation area fraction because of unavailability of pre-monsoon agricultural census data.

2 Reviewer#2

2.1 Comment 1

This study follows on and attempts to rebut that of Mishra et al 2020 on the role of irrigation in India in driving increases in moist heat stress. This is an important topic worthy of further investigation and no study should be considered the last word on the topic. From that point of view I find the study to be relevant and worthy of publication. The main claim of the study is that, in the pre-monsoon season, (Apr-May) irrigation is not much practiced, so that any trend that is ascribed to irrigation, or modeling that supports it, must be wrong. This is certainly an idea worth following up on, since Mishra et al did not provide observational justification for the specific irrigation numbers that they employed in their modeling. But, this also means to make the point convincingly, the irrigation data themselves become a major bone of contention and should be extensively documented and supported.

Response

We thank Reviewer#2 for an encouraging feedback. The irrigation data source and its preparation have been explained in detail in Supplementary Note 1.

2.2 Comment 2

One problem with this study is that the main results of the study hinge completely on an irrigation data set that is poorly described, buried in supplemental material, and unverified. There are several other problems which I will describe more below and I will also suggest some solutions to these problems. In the end, I see value in this study, but I think extensive rethinking, rewriting, and some new experiments and recalculation are in order.

Response

We agree on the importance of the irrigation dataset for this study. Our objective is to highlight the same and demonstrate the importance of region-specific data and a region-specific model while investing the feedback of irrigation. We have now explained the irrigation dataset used in AGR simulation in Supplementary Note 1, describing the different variables required for the input. We have added some near-surface climate variable (Supplementary Figure 11-14) sensitivity to parameterization schemes to make the results robust with the description in Supplementary Note 3.

2.3 Comment 3

First, let me start with what I think we all agree on. Moist heat stress in the region probably arises from a complex interplay of local and regional hydrology including moisture transport and irrigation, as well as inputs of radiation, all of which are impacted by atmospheric circulation, rainfall, clouds and aerosols. Indeed, this study agrees in important ways with the Mishra et al study. Their MOD experiment, in which water is added as need to maintain saturation produces results that are very similar to those in the Mishra study. So, if enough water is added, I think that both studies reach the same answer. Major cooling, increases in humidity, reductions in boundary layer height. Indeed the MOD simulations in this study look surprisingly similar to those in Mishra et al. The main role of WRF/CLM simulations in the Mishra et al paper was to as a sensitivity study to demonstrate the physical mechanisms by which irrigation might cause enhanced moist heat stress, and in that sense this study does not change that picture. While not discussed much in the Mishra et al study, a series of sensitivity tests to varying irrigation were actually carried out and discussed in supplemental material (Figure S11 and S13) and the conclusions for those sensitivity tests were that there was a simple linear relationship between the degree of irrigation and the impact on relevant parameters.

Response

The reviewer is correct that increasing moist heat stress in the region results from various forcings in the region. Further, it is important to understand the contribution of each factors influencing the moist heat stress in the region. However, Mishra et al.¹ have solely attributed the increasing moist heat stress to irrigation. While the sensitivity study of irrigation to near-surface meteorology has been established in the past in the other studies¹⁰, Mishra et al. have proposed the theory of reduction in boundary layer height increasing moist heat stress in the Indo-Gangetic Plain. We found that the results obtained in Mishra et al. are driven by model artifacts rather than real irrigation practices and scenarios. The present study shows that the realistic irrigation application has limited influences on pre-monsoon heat stress in the Gangetic basin. Aerosols and remote moisture transport drive the observed characteristics of land surface cooling and increased humidity. This is established using both physics-based models and statistical analysis.

Now, we have modified couple of sentences “*Hence, the model-estimated irrigation volumes lead to very high feedback on near-surface climate as compared to changes in observed*

records of land surface temperature and wet-bulb temperature. Therefore, attribution of decreasing land surface temperature^{1,2} and rising wet-bulb temperature^{1,17,20} over the Indo-Gangetic Plain to irrigation¹ alone based on over-estimated irrigation overlook contribution from other factors like aerosol loading¹⁶ and remote moisture transport as important as irrigation.” (Lines 57-62)

2.4 Comment 4

Thus, the only really important issue that is addressed in this new paper is whether the AGR irrigation scenario is so much different, and so much closer to reality, that it somehow invalidates the argument that part of the trend in wet bulb temperature ascribed in the Mishra et al study was due to irrigation. It is clear from Figure 1b that (in their estimation) irrigation in UP and Bihar is quite close to zero in the pre-monsoon season. So it is actually not surprising that their results resemble the weaker irrigation states studied in Mishra et al (Figure S13). The justification given in the text for this estimation is, frankly, unclear (245-275). Especially how the irrigation is distributed specifically during the Apr-May window (there is no description of how that is done and the source website (<https://aps.dac.gov.in/LUS/Public/Reports.aspx>) does not have that information in any clear way. I'm not saying this was done improperly, but given that the entire paper depends on it, I can not support publication until this element is transparent and validated.

Response

We agree that pre-monsoon irrigation representation need to be clear for this study. Our intention is also to highlight the contrast in pre-monsoon (Apr-May) and monsoon irrigation (Jun-Sep), requiring region-specific data and a region-specific model while investigating the irrigation feedback on near-surface climate. The information on distribution of irrigation during April-May is also reported in FAO irrigated crop calendar database⁶ for India at monthly scale (<https://www.fao.org/aquastat/en/databases/crop-calendar/>), where April-May is shown to have zero or least percentage of crop irrigated area for different crops (FAO irrigation crop calendar database along with census-based irrigation data worksheet are shared as data source with revised manuscript).

The irrigation dataset prescribed to the model here contains two important information: irrigated area fraction and irrigation wateruse. The irrigation area fraction used in Mishra et al.¹ is FAO irrigation map which represents the annual irrigation area fraction. Similarly, irrigation

wateruse in Mishra et al.¹ is prescribed as precipitation when the root-zone soil moisture fell below the field capacity as quoted below from their method section.

"The fractional area under irrigation from the Food and Agriculture Organization irrigation map was obtained to implement the irrigation scheme in the Noah land surface model coupled with WRF. Irrigation was applied as precipitation when the root-zone soil moisture fell below the field capacity"

Here, we have corrected these two misrepresentation of irrigation during pre-monsoon season using census-based irrigation data in AGR experiment over Indo-Gangetic Plain. The irrigation area fraction is calculated based on area under the crop and crop area irrigated data provided at district level and irrigation wateruse is estimated by using irrigation water required given by Fishman et al.²⁴ for each crop that is produced during the pre-monsoon season.

We have now cited the relevant reports in *"In India, large-scale irrigation is observed only during the monsoon and post-monsoon seasons: Kharif and Rabi, extending from June to September and November to February, respectively. The rest of the months, March to May is usually hot and dry without extensive agricultural activities because of cropping pattern^{5,6} and government policies related to groundwater conservation⁷"* (Lines 44-48) and provided detailed explanation of the irrigation data source and irrigation data input for the AGR experiment in Supplementary Note 1. In addition Supplementary Table 1 provides information on cropping activities.

In addition to irrigation area fraction in Figure 1b,c, we have added irrigation wateruse (mm/day) for paddy and non-paddy crops for pre-monsoon and monsoon season in Supplementary Figure 3. Our irrigation dataset corresponds to the remotely sensed EVI, ET and LAI during pre-monsoon season shown in Figure 1d,f and Supplementary Figure 4, which validates that pre-monsoon have limited irrigation.

2.5 Comment 5

I further raise this concern because of this 269-271: "The average irrigation water withdrawal data from Huang et al.²⁹ and the irrigated area fractions from Food and Agricultural Organisation (FAO) Global Map of Irrigated Areas (GMIA)²⁷ are used over the model domain where agricultural census estimates are not available." The authors have used a completely different dataset everywhere else in India, and it is very clear in the regions just North and West of UP, that the Huang et al dataset has strong

irrigation. There is a noticeable, and unphysical break in all the AGR results across this line between the author's derived water use (UP&Bihar) and the rest of the map (this shows up in Figure 2b, and 2C suggesting that MOD is not actually in grave disagreement with the Huang et al dataset, which is partially used in AGR). This shows up through out Figure 3 as well (and the rest of the paper). Huang et al, from an independent study, which is used outside of 2 states in this study, appears to be not far off from MOD, but completely different than the irrigation results the authors have derived (from opaque methods) in UP and Bihar. To improve this paper, a set of WRF simulations with Huang et al irrigation needs to be performed for comparison and a separate section comparing the authors' estimates in UP and Bihar against those in Huang et al (and presumably a statement why theirs are better).

Response

We thank reviewer for noting the similarity between Huang et al.¹³ dataset and MOD experimental results. We now have performed a new set (HNG experiment) of pre-monsoon (Feb-May) simulations using Huang et al. irrigation data for 2004, 2008, 2012, and 2016, with two initial months taken as spin-off time at the horizontal spatial resolution of 30km and 30 vertical levels using coupled WRF-CLM model.

The result shows that the irrigation withdrawal data from Huang et al. over Indo-Gangetic Plain overestimates the irrigation amount and hence higher evaporative cooling and moistening (Figure 3 and Supplementary Figure 9); however, the overestimation is still lower than Mishra et al.¹ results over the same region. One of the reasons behind the over-estimation of irrigation impact may be the estimation of a higher irrigation requirement during dry April-May months in water models used in Huang et al. study, failing to acknowledge the pre-monsoon irrigation practice in Indo-Gangetic Plain.

We have now added the dry heat stress and moist heat stress response to Huang et al.¹³ irrigation withdrawal data in Supplementary Figure 9 with simulation details explained in Supplementary Note 2. In addition, we have added a few sentences in separate paragraph.

"Moreover, the results from another set of simulation using Huang et al.¹³ monthly irrigation withdrawal data (HNG) all over India for four years (as described in Supplementary Note 2) shows high feedback to meteorological variables (Supplementary Figure 9) similar to the MOD experiment over the Indo-Gangetic Plain. The most probable reason is the tendency of water models to estimate higher irrigation water for dry soil conditions over annual irrigation area

fraction given by GMIA data during pre-monsoon season. The agricultural census-based irrigation volume prescribed to the model (ARG) overcomes this drawback and shows the actual influence of irrigation." (Lines 161-168)

Also, the contrasting response over UP and Bihar and other parts of Gangetic Plain (Punjab and Haryana) is due to the different irrigation data (census-based irrigation data in UP-Bihar and Huang et al. irrigation data in other parts) used in the respective region. Now, we have updated our region of interest to the Indo-Gangetic Plain (UP, Bihar, Haryana, Punjab and Rajasthan) based on the updated crop calendar (Supplementary Table 1) from Indian Agricultural statistics and crop production data for the pre-monsoon/summer season and prepared the required irrigation datasets as described in Supplementary Note 1. With the updated irrigation dataset, the LST and wet-bulb temperature and other meteorological variables response to irrigation during April-May are consistent over Indo-Gangetic Plain except over a sugarcane-growing areas where higher cooling and moistening is observed since sugarcane is a water-intensive crop. Now, the Figure 2 and Figure 3 shows consistent irrigation feedback on different variables across the different parts of Indo-Gangetic Plain.

2.6 Comment 6

In addition to that, which is my main scientific concern, there are several technical issues that should be addressed, and these will require further simulations.

First, it is not just irrigation that is the issue, it is the soil moisture. These simulations are begun and spun-up for a relatively brief time. What is the initial soil moisture distribution? In the real world, irrigation signals may persist within soil moisture for month's anomalies. Unless the soil moisture is initialized as saturated and then spun up, then, in the absence of rain and (weak specified) irrigation then the soil moisture will be much drier in the AGR simulations than is perhaps realistic. A sensitivity tests to this (by beginning with wet soils) would help.

Response

We have taken two months of spin-up time (Feb-Mar) to reach physical equilibrium. Several studies²⁵⁻²⁷ over Indian region focussing on land-atmosphere interaction have considered one month spin-up time for performing different experiment in WRF-CLM. Now, we have added a sentence citing other studies adopting similar spin-up time.

“Similar spin-up period is considered in other studies²⁵⁻²⁷ for performing experiments over Indian region in WRF-CLM.” (Line 281-283)

Further, we agree with the reviewer that the soil moisture-temperature feedback influences the near-surface climate. It is also true that irrigation signals may persist for a longer time scale; however, in reality, irrigation is minimal from February to May, and soil moisture is well below saturation level and even field capacity. Here, Figure R1 shows that ESA CCI²⁸ observed soil moisture (0-5cm) is a small fraction (30-40%) of saturated soil moisture over the Indo-Gangetic Plain, and the climatology of soil moisture (Figure R2) reveals that observed soil moisture(0-5cm) is around half or less than half of the saturation level. So, soil moisture initialized as saturated will be unrealistic over this region.

Here, the saturation soil moisture is estimated using formula used in Community Land model²⁹ (CLM4).

$$\text{Sat_SM} = 0.489 - 0.00126 * \text{sand_frac}$$

$$\text{Sat_SM} = (1 - \text{om_frac}) * \text{watsat} + \text{om_watsat} * \text{om_frac}$$

Where, sand_frac is the fraction of sand and om_frac is the fraction of organic matter for different soil type.

Figure R1 Percentage of observed soil moisture(0-5cm) with respect to saturated soil moisture(0-5cm).

Figure R2 Climatology of observed soil moisture with saturated soil moisture value

2.7 Comment 7

Second, as far as I can tell wet bulb is being calculated from averaged quantities, not for the original hourly (or 3-hourly) output. Since this is a highly non-linear quantity that varies strongly over the diurnal cycle, daily or even 12 hourly means of T, RH, etc are not sufficient. Wet bulb should be calculated at high resolution (as it was in the Mishra et al study). 1-2° Tw errors can arise from this and the offset may be strongly impacted by exactly the boundary layer process that are crucial for this study.

Response

We thank the reviewer for the suggestion. We now have updated our calculation accordingly and modified our method section as "*The simulated meteorological variables (mean temperature, maximum temperature, minimum temperature, specific humidity, relative humidity, pressure, heat fluxes, and wet-bulb temperature) at an hourly scale is converted to daily scale and then averaged over the simulation period for each set of the experiment*". (Lines 321-324)

2.8 Comment 8

Third, and in a similar vein, the Stull approximation to wet bulb temperature is substantially inaccurate away from typical mid-latitude conditions and should not be used in this study. Many more accurate techniques exist for this computation and freely available packages can be used to do it. This should be done.

Response

We have now used the wet-bulb temperature package provided in NCL and updated in the method section as *"The hourly wet-bulb temperature is calculated using hourly mean temperature, hourly dew-point temperature, and hourly surface pressure using the iterative procedure described by Stipanuk³⁰ available in NCL"* (Line 324-326).

3 Reviewer#3

3.1 Comment 1

This study explores the over-estimation of irrigation impacts on heat stress over the Indian regions. They use the WRF model combined with the satellite dataset to demonstrate previous studies might significantly over-estimate (overemphasize) the irrigation's cooling impacts. They further argue that aerosol might play a more prominent role in surface cooling than surface irrigation. The authors raise one critical point of the seasonality issue: the irrigation season and dry season do not coincide. The previous studies may misinterpret; further, the model's irrigation scheme may over-estimate the irrigation water amount, thus, over-estimating the cooling effect from irrigation. In general, this study provides some new information to the community about how we can better represent the irrigation impact in the model and consider both temporal impacts of irrigation and the local mean climate (whether the irrigation season is the same as the dry season).

Response

We thank Reviewer#3 for understanding the seasonality issue of irrigation and heat extremes in the Indo-Gangetic Plain, and also for providing an overall encouraging feedback.

3.2 Comment 2

The results look interesting, and this study provides some valuable information for the community. However, this paper needs to include more discussions on the uncertainties and scientific questions exploration, and some comments need to be addressed. These are outlined below.

Response

We totally agree with the reviewer and have addressed the comments with new simulations and sensitivity analysis explained in Supplementary Notes 2 and 3.

3.3 Comment 3

1. Besides showing the annual, pre- and post-monsoon irrigation fraction, how about the irrigation water? During the monsoon season, the irrigation amount should be minimal as well due to a large amount of rainfall? Or it's the other way around? So, only showing

the irrigation fraction is not enough; please also demonstrate and discuss the irrigation water amount's temporal variations.

Response

Pre-monsoon season (April and May) is a non-cropping season with limited irrigated area and irrigation water use over the Indo-Gangetic plain. Limited crops are grown during pre-monsoon season. This information is also reported in FAO irrigated crop calendar database⁶ for India at monthly scale (<https://www.fao.org/aquastat/en/databases/crop-calendar/>), where April-May is shown to have zero or least percentage of crop irrigated area for different crops (FAO irrigation crop calendar database along with census-based irrigation data worksheet are shared as data source with revised manuscript).

The irrigation water use for pre-monsoon and monsoon for paddy and non-paddy crops are now shown in Supplementary Figure 3. The irrigation water for different crops is taken from Fishman et al.²⁴, as shown in Supplementary Table 1. The temporal variation in irrigation water use is discussed as "*In addition, the irrigation water requirement for paddy and non-paddy crops taken from Fishman et al.²⁴ shows lower irrigation water for both paddy and non-paddy crops during pre-monsoon season than monsoon season (Supplementary Fig. 3)*". (Line 94-97).

Since the major cropping season (Kharif) coincides with monsoon season: Jun-Sep, there is extensive irrigation (irrigated area fraction of above 0.8) during monsoon season as already reported by different studies^{14,26}. In this period, irrigation water use is also high as mostly flood irrigation is practiced over paddy fields in the Indo-Gangetic Plain. We have now added some sentences in the introduction section and cited the relevant reports to bring clarity on this.

"In India, large-scale irrigation is observed only during the monsoon and post-monsoon seasons: Kharif and Rabi, extending from June to September and November to February, respectively. The rest of the months, March to May is usually hot and dry without extensive agricultural activities because of cropping pattern^{5,6} and government policies related to groundwater conservation⁷." (Lines 44-48)

3.4 Comment 4

2. The authors have also argued that limited role in hot extremes during fewer irrigation activities in the pre-monsoon season. More analysis on the irrigation activities is needed to be more convincing.

Response

The preparation of irrigation data source and irrigation data input for AGR experiment have been explained in detail in Supplementary Note 1. Further, Supplementary Table 1 provides information on pre-monsoon cropping activities. The irrigation wateruse for pre-monsoon and monsoon for paddy and non-paddy crops are now shown in Supplementary Figure 3. In addition, the irrigation data worksheet along with FAO irrigation crop calender for India has been attached to the supplementary file. We are happy to carry out further analysis on the irrigation activities if the reviewer has any particular additional idea in mind; from the comment, currently, it is not clear what further analysis is expected.

3.5 Comment 5

3. Figure 4c is interesting, and can the authors also show the scatter plot for the whole India region?

Response

The daytime LST relation with AOD for the whole India has been included as Supplementary Figure 18a. The scatter plot with correlation coefficient shows a significant weak positive correlation, $r = 0.18$. The contrasting relation between these two variables for Indo-Gangetic Plain and India is because of high aerosol loading over Indo-Gangetic Plain compared to other parts of India and different hydro-climatic conditions prevalent in different regions. We have added a sentence in the main text *"However, a weak positive linear relationship is witnessed between AOD and daytime LST over the Indian region as AOD and LST observed over the rest of India is different from the Indo-Gangetic plain (Supplementary Fig. 18a)."* (Line 234-236).

3.6 Comment 6

Validation: Extended data figure 4 may not be enough for the model-observation validation. How about showing the pattern correlation?

Response

For the validation purpose, Supplementary Figure 5 shows the difference between model and IMD²² observed data for daily mean temperature instead of validating with ERA5 data. The sentence is updated in the main text as: *"Model simulated 2m-air temperature from three different parameterization schemes are compared with those from Indian Meteorological Department (IMD) observed data²² to assess model skill over India. The WRF-CLM4 model with MYNN3-WSM6-Grell 3D combination shows the least difference between observed and*

simulated result simulates over Indo-Gangetic Plain for April-May during 2004 and 2006 (Supplementary Fig. 5)." (Lines 129-134)

We have now added *"Moreover, the pattern correlation for two variables: daily mean temperature and daily maximum temperature between model simulated and observed data showed good correlation between them over the Indo-Gangetic Plain (Supplementary Fig. 6)".* (Line 134-137)

3.7 Comment 7

Does the WRF include both aerosol direct and indirect effects? Using WRF to simulate the aerosol's effect on such surface cooling tendency will be a nice add-on. Then, the authors can further discuss the aerosol's direct and indirect effect on the surface cooling here.

Response

The present simulation uses a coupled WRF-CLM model without considering direct and indirect aerosol effects. The quantification of the influence of aerosol loading on heat stress extremes requires WRF-Chem modeling, which is beyond the objectives of the present study. One recent study using the WRF-Chem model by Dey et al.¹⁶ have shown that high aerosol loading in India could mask the heat stress (quantified by the Wet Bulb Globe Temperature, WBGT) by 0.3-1.5°C in 2010. We now have cited this study in *"Model-driven studies have shown the direct and indirect effects of aerosol-radiation interaction on lowering the temperature²¹ and increasing the relative humidity¹⁶".* (Line 228-230)

3.8 Comment 8

WRF provides various PBL and convection parameterizations, so will the different parameterizations affect the results shown here? Some sensitivity tests for this regard can make the results more robust.

Response

The sensitivity of result to a different combination of parameterization schemes is carried out and explained in Supplementary Note 3. The results are shown in Supplementary Figure 11-14. The different parameterization schemes used do not largely affect the output suggesting the robust result.

We now have added “*Moreover, the sensitivity test of daily mean temperature (Supplementary Fig. 11), daily maximum temperature (Supplementary Fig. 12), and wet-bulb temperature (Supplementary Fig. 13) response to model-estimated irrigation and agricultural census-based irrigation for a different combination of parameterization schemes (Supplementary Note 3) show results are quite robust. The error bar diagram (Supplementary Fig. 14) shows that the agricultural census-based irrigation has a similar influence with three parameterization combinations*”. (Lines 189-195)

3.9 Comment 9

Using the reanalysis data as the boundary condition to force the WRF model may lack the remote impacts, such as the whole monsoon strength? Some studies already mentioned that the critical role of India's massive irrigation may affect India's monsoon strength. Will the weak signal in this study come from the lack of remote impacts or large-scale response from Indian irrigation? Or, the further question will be: whether the WRF platform's boundary condition affects the results in this study?

Response

The simulation is carried out from Feb-May, during which monsoon is not active in India. The expected onset date of the monsoon is 1 June in the Southern part of India. Some studies^{14,26} have explored the irrigation impacts on India's monsoon strength during major cropping season (June-Sep) when irrigation is high; however, here, we are interested in pre-monsoon irrigation during April-May and its influence on heat stress. Therefore, the WRF platform's boundary condition will not affect the results presented here.

3.10 Comment 10

8. The authors also argue that "However, the significant irrigation expansion is observed during monsoon and high aerosol loading measured during dry pre-monsoon, which depicts that cooling trend during pre-monsoon is associated strongly with high aerosol loading than limited pre-monsoon irrigation" based on Shukla, S. P., Puma, M. J. & Cook, B. I. The response of the South Asian Summer Monsoon circulation to intensified irrigation in global climate model simulations. *Clim. Dyn.* 42, 21–36 (2014). Not sure Shukla et al., 2014 can show the significant irrigation expansion? What does it mean?

Response

Shukla et al.¹⁴ have shown the comparison of pre-monsoon and monsoon irrigation water use from 1960-2000 in Figure 1 (reproduced below). While the pre-monsoon irrigation water use is very limited and has not increased significantly over a period of time, significant irrigation expansion has been observed during the monsoon season as seen in the figure. Hence, the cooling and moistening trend must be associated with other forcing like aerosol loading in addition to irrigation. We have now slightly modified the sentence for clarity “*However, a significant irrigation expansion is observed only during monsoon¹⁴, and aerosol loading increasing since 1980s is high during pre-monsoon¹⁵, which depicts that the cooling and moistening trend that is observed during pre-monsoon is associated with other factors like high aerosol loading¹⁶ than limited pre-monsoon irrigation over Gangetic Plain (Uttar Pradesh and Bihar)*”. (Lines 249-253)

Fig. 1 Irrigation water (mm/day) added over the Indian sub-continent (5°–30°N, 60°–95°E) in ModelE for Monsoon Season (*red*) and pre-monsoon dry season (*blue*) between 1960 and 2001

4 References

1. Mishra, V. *et al.* Moist heat stress extremes in India enhanced by irrigation. *Nat. Geosci.* **13**, 722–728 (2020).
2. Joshi, M. K., Rai, A., Kulkarni, A. & Kucharski, F. Assessing Changes in Characteristics of Hot Extremes Over India in a Warming Environment and their Driving Mechanisms. *Sci. Rep.* **10**, 1–14 (2020).
3. Rohini, P., Rajeevan, M. & Srivastava, A. K. On the Variability and Increasing Trends of Heat Waves over India. *Sci. Rep.* **6**, 1–9 (2016).
4. Pai, D. S., Nair, S. A. & Ramanathan, A. N. Long term climatology and trends of heat waves over India during the recent 50 years (1961-2010). *Mausam* **64**, 585–604 (2013).
5. Government of India. *Agricultural Statistics At a Glance 2020*. Government of India Ministry of Agriculture Department of Agriculture and Cooperation Directorate of Economics and Statistics (2020).
6. FAO. AQUASTAT - Global Map of Irrigated Areas. *Food and Agriculture Organization of the United Nations (FAO)* (2021). Available at: <https://www.fao.org/aquastat/en/databases/crop-calendar/>. (Accessed: 25th December 2021)
7. Singh, K. Act to Save Groundwater in Punjab: Its Impact on Water Table, Electricity Subsidy and Environment. *Agric. Econ. Res. Rev.* **22**, 365–386 (2009).
8. Thiery, W. *et al.* Warming of hot extremes alleviated by expanding irrigation. *Nat. Commun.* **11**, 1–7 (2020).
9. Thiery, W. *et al.* Present-day irrigation mitigates heat extremes. *J. Geophys. Res.* **122**, 1403–1422 (2017).
10. Mathur, R. & AchutaRao, K. A modelling exploration of the sensitivity of the India's climate to irrigation. *Clim. Dyn.* **54**, 1851–1872 (2020).
11. Shastri, H., Barik, B., Ghosh, S., Venkataraman, C. & Sadavarte, P. Flip flop of Day-night and Summer-Winter Surface Urban Heat Island Intensity in India. *Sci. Rep.* **7**, 40178 (2017).
12. Miralles, D. G., Teuling, A. J., Van Heerwaarden, C. C. & De Arellano, J. V. G. Mega-

- heatwave temperatures due to combined soil desiccation and atmospheric heat accumulation. *Nat. Geosci.* **7**, 345–349 (2014).
13. Huang, Z. *et al.* Reconstruction of global gridded monthly sectoral water withdrawals for 1971-2010 and analysis of their spatiotemporal patterns. *Hydrol. Earth Syst. Sci.* **22**, 2117–2133 (2018).
 14. Shukla, S. P., Puma, M. J. & Cook, B. I. The response of the South Asian Summer Monsoon circulation to intensified irrigation in global climate model simulations. *Clim. Dyn.* **42**, 21–36 (2014).
 15. David, L. M. *et al.* Aerosol Optical Depth Over India. *J. Geophys. Res. Atmos.* **123**, 3688–3703 (2018).
 16. Dey, S., Choudhary, R. K., Upadhyay, A. & Dash, S. K. Aerosol-modulated heat stress in the present and future climate of India. *Environ. Res. Lett.* **16**, 124022 (2021).
 17. Oldenborgh, G. J. Van *et al.* Extreme heat in India and anthropogenic climate change. 365–381 (2018).
 18. Oldenborgh, G. J. V. *et al.* Extreme heat in India and anthropogenic climate change. *Nat. Hazards Earth Syst. Sci.* **18**, 365–381 (2018).
 19. Siebert, S., Henrich, V., Frenken, K. & Burke, J. Update of the digital global map of irrigation areas to version 5. *Rheinische Friedrich-Wilhelms-Universituy, Bonn, Ger. Food Agric. Organ. United Nations, Rome, Italy* 171 (2013).
 20. Im, E. S., Pal, J. S. & Eltahir, E. A. B. Deadly heat waves projected in the densely populated agricultural regions of South Asia. *Sci. Adv.* **3**, 1–8 (2017).
 21. Bharali, C., Nair, V. S., Chutia, L. & Babu, S. S. Modeling of the Effects of Wintertime Aerosols on Boundary Layer Properties Over the Indo Gangetic Plain. *J. Geophys. Res. Atmos.* **124**, 4141–4157 (2019).
 22. Srivastava, A. K., Rajeevan, M. & Kshirsagar, S. R. Development of a high resolution daily gridded temperature data set (1969 – 2005) for the Indian region. *Atmos. Sci. Lett.* **10**, 249–254 (2009).
 23. Stipanuk, G. S. Algorithms for Generating a Skew-T, Log P Diagram and Computing Selected Meteorological Quantities. (1973).

24. Fishman, R., Devineni, N. & Raman, S. Can improved agricultural water use efficiency save India's groundwater? *Environ. Res. Lett.* **10**, (2015).
25. Devanand, A., Roxy, M. K. & Ghosh, S. Coupled Land-Atmosphere Regional Model Reduces Dry Bias in Indian Summer Monsoon Rainfall Simulated by CFSv2. *Geophys. Res. Lett.* **45**, 2476–2486 (2018).
26. Devanand, A., Huang, M., Ashfaq, M., Barik, B. & Ghosh, S. Choice of Irrigation Water Management Practice Affects Indian Summer Monsoon Rainfall and Its Extremes. *Geophys. Res. Lett.* **46**, 9126–9135 (2019).
27. Paul, S. *et al.* Weakening of Indian Summer Monsoon Rainfall due to Changes in Land Use Land Cover. *Sci. Rep.* **6**, 1–10 (2016).
28. Gruber, A., Scanlon, T., Schalie, R. Van Der, Wagner, W. & Dorigo, W. Evolution of the ESA CCI Soil Moisture climate data records and their underlying merging methodology. 717–739 (2019).
29. Oleson, K. W. *et al.* Technical Description of version 4.0 of the Community Land Model (CLM). *NCAR/TN-478+STR NCAR Tech. NOTE* (2010).
30. Stipanuk, G. Algorithms for Generating a SKEW-T, log p Diagram and Computing Selected Meteorological Quantities. 1–15 (1973).

Reviewers' Comments:

Reviewer #1:

None

Reviewer #2:

Remarks to the Author:

The revised manuscript is much improved compared to the earlier version and I appreciate the efforts the authors have made. Given the current heat waves in the region, this paper is highly topical and relevant. There remains a basic problem with how the study is framed and it is persistent through the paper and in the response to reviewer comments. The authors incorrectly summarize and characterize the methodology and conclusions of the Mishra paper, which unfortunately undermines aspects of the major conclusions of this study.

The root issue here is that in this study, the authors incorrectly characterize the WRF simulations in the Mishra study as aiming to explain in all realism the observed trends in temperature, humidity, wet bulb and boundary layer height. The WRF simulations were idealized sensitivity studies to determine of irrigation could drive the kinds of changes observed. I agree that the irrigation used was larger than is observed (I appreciate the new, better description of this dataset in the revised version), but so too was the magnitude of the response in the meteorological quantities. The Mishra paper was intended to identify trends in these variables and then show that they can be explained by irrigation. Please compare figure 4d-f and Figure 2d-f (and please note the doubling in the scale range in T_w between 4f and 2f). As shown in Figure S13 in the Mishra paper changes in T , boundary layer height etc are rather simple linear functions of irrigation and this is the main idea concept in the paper.

The statement in this manuscript is incorrect and mischaracterizes the results in Mishra, "Further, irrigation prescribed in the AGR experiment does not substantially reduce the Planetary Boundary Layer (PBL) height compared to the MOD experiment (Supplementary Fig. 17a,b), negating the mechanism of increasing moisture heat due to reduction in PBL height as claimed in earlier study²²." Looking at Figure S14, the observed trend in BLH the order of magnitude of change that Mishra is trying to explain is ~50meters an amount that can easily be encompassed with the smaller amounts of irrigation indicated in the present study. Thus Figure S17b confirms the mechanism discussed in Mishra and the results are similar to comparable to those in S13 in Mishra.

This mischaracterization of the results of the Mishra study permeates and unfortunately undermines this study, even beginning in the abstract

"we show that previously reported irrigation effects on heat stress in the region during the pre-monsoon season are 4.9 times overestimated due to the non-consideration of seasonal variations in irrigation application."

This number arises from misunderstanding the Mishra paper, which has two parts: one is based on observations and correlations and the other is a modeling sensitivity study. Or as they put it "Having empirically demonstrated a plausible role for irrigation in increasing moist heat stress from observations, we employed modelling to provide causal attribution."

As has been established in the review process, when the authors in the current study use similar irrigation numbers in their WRF simulations as in the Mishra study, they get the same result. Mishra et al never said that the amount of one value of irrigation in their study was "correct", in fact they used a full range of values. They only concluded that it there was a robust and strong, nearly-linear dependence of relevant meteorological quantities on irrigation, they never claimed any particular irrigation scenario matched observations. Clearly all the patterns in Figure 3 are 2-5 times more amplified than in observations.

This sort of mischaracterization comes up in several places, for example again here:

"Extreme moist heat conditions represented by Tw_95 have increased by 0.22°C over Indo-Gangetic Plain with irrigation application, which is lower than the previous estimates of 3-6°C shown in model-driven study²² (Supplementary Fig. 15b)."

Again, 3-6°C is greater than the observed signal in Mishra et al (Figure 2); the WRF simulations with "irrigation on" were never intended for one-to-one correspondence with reality—these were sensitivity tests. Looking at Figure S13 in Mishra et al and the corresponding simulations in this new manuscript and the results are very similar.

The main result of this study is to highlight that irrigation is on the low-end of the WRF simulations in Mishra et al. and therefore the magnitude of induced changes may be small enough to be dominated by other processes (aerosols, advection, etc). That is a 100% interesting and publishable study. I encourage revisions that focus on those results which are supported by their efforts (which are very similar in their outputs to those of Mishra et al, it is just that they believe that the inputs lie at the low end of those considered in Mishra).

Reviewer #3:

Remarks to the Author:

The authors have addressed most of my comments, and I am satisfied with their responses. While I appreciate their effort in calculating the pattern correlation, the sentence they used to describe the similarity.

The authors added a new sentence: "...daily mean temperature and daily maximum temperature between model simulated and observed data showed good correlation" should be more specific.

They need to be specific on what "good" means in the revised manuscript.

Limited influence of irrigation on pre-monsoon heat stress in the Indo-Gangetic Plain

Manuscript No: NCOMMS-21-32895B

Response to Reviewers

We thank the editorial board and the reviewers for their insightful and encouraging comments. Below we provide a point by-point response to the reviewers' comments and details of the actions taken. Figures, tables, line and page numbers mentioned in this document refer to the revised manuscript, unless specified otherwise. Reviewer comments are shown in **bold**. Author responses are shown in plain text.

1 Reviewer#2

1.1 Comment 1

The revised manuscript is much improved compared to the earlier version and I appreciate the efforts the authors have made. Given the current heat waves in the region, this paper is highly topical and relevant. There remains a basic problem with how the study is framed and it is persistent through the paper and in the response to reviewer comments. The authors incorrectly summarize and characterize the methodology and conclusions of the Mishra paper, which unfortunately undermines aspects of the major conclusions of this study.

Response

We thank the reviewer for understanding the importance and relevance of the study. It is imperative to correctly characterise the role of irrigation on pre-monsoon heat stress in India for policymakers to address the human and public health effects of heat stress. Here, we have intended to show that models/methods simulating irrigation for other parts of the world are not suitable for application in the Indian region. Our intention is not to criticize Mishra et al., (2020), highlighting their limitations, but in a larger context, we wanted to show the inadequacy of the state-of-the-art modeling framework in simulating the Indian human-natural climate system. Hence, model-driven hypothesis testing with such a global framework may not always result in a realistic conclusion. The present case is an example where we have tried to address the model artifact in prescribing irrigation over the Indo-Gangetic plain during the pre-monsoon season and compared it to previous studies to show the improvements. We have now described this in conclusion (Line 273-275).

"The model-driven hypothesis testing with state of the art modelling framework is found to be inadequate in simulating Indian human-natural climate system resulting in non-realistic conclusion."

1.2 Comment 2

The root issue here is that in this study, the authors incorrectly characterize the WRF simulations in the Mishra study as aiming to explain in all realism the observed trends in temperature, humidity, wet bulb and boundary layer height. The WRF simulations were idealized sensitivity studies to determine of irrigation could drive the kinds of changes observed. I agree that the irrigation used was larger than is observed (I appreciate the new, better description of this dataset in the revised version), but so too was the magnitude of the response in the meteorological quantities. The Mishra paper was intended to identify trends in these variables and then show that they can be explained by irrigation. Please compare figure 4d-f and Figure 2d-f (and please note the doubling in the scale range in Tw between 4f and 2f). As shown in Figure S13 in the Mishra paper changes in T, boundary layer height etc are rather simple linear functions of irrigation and this is the main idea concept in the paper.

Response

We agree with the reviewer that Mishra et al., (2020) have shown a simple linear relationship between irrigation and changes in specific humidity and temperature through sensitivity

experiments based on different thresholds of irrigation fractions, as shown in Figure S13. We have now stated the same Line 32-34.

"Multiple studies also showed the magnitude of cooling is a linear function of the volume of irrigation through sensitivity analyses." (Line 32-34)

As mentioned by the reviewer, Mishra et al. (2020) concluded that the recent trends of wet bulb temperature and humidity can be explained by pre-monsoon season irrigation. We respectfully disagree with this conclusion and showed that the model artifacts resulted in such a conclusion in Mishra et al. (2020). In the manuscript, we highlighted the same. We have further clarified in Line 278.

1.3 Comment 3

The statement in this manuscript is incorrect and mischaracterizes the results in Mishra, "Further, irrigation prescribed in the AGR experiment does not substantially reduce the Planetary Boundary Layer (PBL) height compared to the MOD experiment (Supplementary Fig. 17a,b), negating the mechanism of increasing moisture heat due to reduction in PBL height as claimed in earlier study22." Looking at Figure S14, the observed trend in BLH the order of magnitude of change that Mishra is trying to explain is ~50meters an amount that can with easily be encompassed the smaller amounts of irrigation indicated in the present study. Thus Figure S17b confirms the mechanism discussed in Mishra and the results are similar to comparable to those in S13 in Mishra.

Response

We agree that the statement is incorrect, considering there will be an increase in moisture with a reduction in PBL height irrespective of forcing causing the decrease in PBL height. We have now removed the latter part of the sentence.

"Further, irrigation prescribed in the AGR experiment does not substantially reduce the Planetary Boundary Layer (PBL) height compared to the MOD experiment (Supplementary Fig. 17a,b)." (Lines 218-220)

1.4 Comment 4

This mischaracterization of the results of the Mishra study permeates and unfortunately undermines this study, even beginning in the abstract "we show that previously reported irrigation effects on heat stress in the region during the pre-monsoon season are 4.9 times overestimated due to the non-consideration of seasonal variations in irrigation application." This number arises from misunderstanding the Mishra paper, which has two parts: one is based on observations and correlations and the other is a modeling sensitivity study. Or as they put it "Having empirically demonstrated a plausible role for irrigation in increasing moist heat stress from observations, we employed modelling to provide causal attribution."

Response

We thank the reviewer for the comment.

Simulation based studies to understand the impacts of irrigation on heatwaves overestimate the irrigation amount in the pre-monsoon season in India and hence their impacts. Here, we quantified the overestimation of irrigation impacts on heatwaves in India by the state-of-the-

art globally used regional modeling framework. Such frameworks were used not only by Mishra et al. (2020) but also by Thiery et al. (2017, 2020). Following the reviewer's suggestions, we have now changed the statement to:

"Here, using observed irrigation data and regional climate model simulations, we show that irrigation effects on heat stress during pre-monsoon are 4.9 times over-estimated with model-simulated irrigation as prescribed in previous studies" (Lines 17-20)

1.5 Comment 5

As has been established in the review process, when the authors in the current study use similar irrigation numbers in their WRF simulations as in the Mishra study, they get the same result. Mishra et al never said that the amount of one value of irrigation in their study was "correct", in fact they used a full range of values. They only concluded that it there was a robust and strong, nearly-linear dependence of relevant meteorological quantities on irrigation, they never claimed any particular irrigation scenario matched observations. Clearly all the patterns in Figure 3 are 2-5 times more amplified than in observations. This sort of mischaracterization comes up in several places, for example again here: "Extreme moist heat conditions represented by Tw_95 have increased by 0.22°C over Indo-Gangetic Plain with irrigation application, which is lower than the previous estimates of 3-6°C shown in model-driven study22 (Supplementary Fig. 15b)." Again, 3-6°C is greater than the observed signal in Mishra et al (Figure 2); the WRF simulations with "irrigation on" were never intended for one-to-one correspondence with reality—these were sensitivity tests. Looking at Figure S13 in Mishra et al and the corresponding simulations in this new manuscript and the results are very similar.

Response

Our intention here is to compare the irrigation effects from census-based irrigation data and model-driven irrigation data. The results are also compared with previous studies that have prescribed irrigation using a similar method that is unsuitable for Indian conditions, especially during the pre-monsoon season. The objective here is to show that Indian irrigation practices during pre-monsoon are different from other regions of the world, requiring a land surface model to accommodate field-specific conditions in estimating irrigation amount.

Now, we have added the word 'simulated' for clarity.

"Extreme moist heat conditions represented by Tw_95 have increased by 0.22°C over Indo-Gangetic Plain with irrigation application, which is lower than the previous model simulated estimates of 3-6°C shown in model-driven study²² (Supplementary Fig. 15b)". (Lines 213-216)

1.6 Comment 6

The main result of this study is to highlight that irrigation is on the low-end of the WRF simulations in Mishra et al. and therefore the magnitude of induced changes may be small enough to be dominated by other processes (aerosols, advection, etc). That is a 100% interesting and publishable study. I encourage revisions that focus on those results which are supported by their efforts (which are very similar in their outputs to those of Mishra et al, it is just that they believe that the inputs lie at the low end of those considered in Mishra).

Response

We agree with the reviewer that the magnitude of irrigation cooling is small enough to be dominated by other factors. The IPCC also assessed that the effect of increased greenhouse gas on temperatures extremes is moderated or amplified at local scales by other factors such as irrigation, aerosols, and land-use changes (IPCC Chapter 11, 2021). This was discussed in Oldenborgh et al., (2022). The representation of these local factors is either absent or misrepresented in most of the global climate models for the Indian region. Our effort here is to represent the actual irrigation in the model to understand local factors' role at a regional level. The census-based irrigation data can be on the low-end of WRF sensitivity experiments in Mishra et al. 2020, however, our method is more robust and accurate for estimating the irrigation influence on near-surface climate. Our study also strengthens the importance of having field data for land-atmosphere feedback studies. This is now highlighted in Lines 281-282, 285-286.

2 Reviewer#3

2.1 Comment 1

The authors have addressed most of my comments, and I am satisfied with their responses. While I appreciate their effort in calculating the pattern correlation, the sentence they used to describe the similarity. The authors added a new sentence: "...daily mean temperature and daily maximum temperature between model simulated and observed data showed good correlation" should be more specific. They need to be specific on what "good" means in the revised manuscript.

Response

We have now changed the qualitative sentence to quantitative to show the agreement between simulated and observed data.

"Moreover, the pattern correlation for two variables: daily mean temperature and daily maximum temperature between model simulated and observed data showed a correlation value of greater than 0.6 over the Indo-Gangetic Plain (Supplementary Fig. 6)." (Lines 143-146)

3 References

1. Mishra, V. *et al.* Moist heat stress extremes in India enhanced by irrigation. *Nat. Geosci.* **13**, 722–728 (2020).
2. Thiery, W. *et al.* Warming of hot extremes alleviated by expanding irrigation. *Nat. Commun.* **11**, 1–7 (2020).
3. Thiery, W. *et al.* Present-day irrigation mitigates heat extremes. *J. Geophys. Res.* **122**, 1403–1422 (2017).
4. Oldenborgh, G. *et al.* Attributing and projecting heatwaves is hard: We can do better. *Earth's Future*, 10 (2022).
5. Seneviratne, S.I. *et al.* Weather and Climate Extreme Events in a Changing Climate.

Climate Change 2021: The Physical Science Basis. Contribution of Working Group I to the Sixth Assessment Report of the Intergovernmental Panel on Climate Change. Cambridge University Press, Cambridge, United Kingdom and New York, NY, USA, 1513–1766, (2021)